# Artificial kagome lattices of Shockley surface states patterned by halogen hydrogen-bonded organic frameworks

Ruoting Yin[1,4], Xiang Zhu[1,4], Qiang Fu [1,2], Tianyi Hu [1], Lingyun Wan[1], Yingying Wu[1], Yifan Liang[1], Zhengya Wang[1], Zhen-Lin Qiu[3], Yuan-Zhi Tan [3], Chuanxu Ma [1,2] ✉, Shijing Tan [1,2], Wei Hu [1,2], Bin Li[1,2], Z. F. Wang [1,2], Jinlong Yang [1,2] & Bing Wang [1,2] ✉

Artificial electronic kagome lattices may emerge from electronic potential landscapes using customized structures with exotic supersymmetries, benefiting from the confinement of Shockley surface-state electrons on coinage metals, which offers a flexible approach to realizing intriguing quantum phases of matter that are highly desired but scarce in available kagome materials. Here, we devise a general strategy to construct varieties of electronic kagome lattices by utilizing the on-surface synthesis of halogen hydrogen-bonded organic frameworks (XHOFs). As a proof of concept, we demonstrate three XHOFs on Ag(111) and Au(111) surfaces, which correspondingly deliver regular, breathing, and chiral breathing diatomic-kagome lattices with patterned potential landscapes, showing evident topological edge states at the interfaces. The combination of scanning tunnelling microscopy and noncontact atomic force microscopy, complemented by density functional theory and tight-binding calculations, directly substantiates our method as a reliable and effective way to achieve electronic kagome lattices for engineering quantum states.

Electronic lattices provide a tunable playground to harness quasiparticles for realizing designer topological states and correlated phases of quantum matter[1–4], such as the kagome lattice[5], which can produce interaction-driven fractional quantum Hall states[6], charge ordering[7,8], and topological superconductivity[9,10]. These intriguing properties are dictated by the specific electronic structures[11–14] that host both Dirac cones and flat bands (FBs) in the two-dimensional (2D) kagome lattice, consisting of triangular and hexagonal motifs in a network of corner-sharing triangles. Several kagome lattices with slight changes at the triangles can give rise to additional exotic quantum phenomena, such as the emerging topological edge states[15,16] and high-

order topological corner states[17,18] in the breathing kagome lattice and the presence of two FBs in the diatomic-kagome lattice consisting of two atoms sitting on each kagome lattice site[19,20]. Although thousands of compound crystals are known to have the kagome net in their certain sublattices, only a few compounds have been identified to have electronic properties corresponding to the kagome lattice[9–12] and breathing kagome lattice[13,21,22]. Theoretical calculations show that only approximately 7% of the potential kagome compounds possess electronic properties related to the kagome sublattice because the complicated interactions from the interlayer atoms can significantly alter and even destroy the kagome electronic bands[23]. The 2D nature of

[1]Hefei National Research Center for Physical Sciences at the Microscale and Synergetic Innovation Center of Quantum Information & Quantum Physics, New Cornerstone Science Laboratory, University of Science and Technology of China, Hefei 230026 Anhui, China. [2]Hefei National Laboratory, University of Science and Technology of China, Hefei 230088, China. [3]Collaborative Innovation Center of Chemistry for Energy Materials, State Key Laboratory for Physical Chemistry of Solid Surfaces, and Department of Chemistry, College of Chemistry and Chemical Engineering, Xiamen University, Xiamen 361005, China. [4]These authors contributed equally: Ruoting Yin, Xiang Zhu. ✉e-mail: cxma85@ustc.edu.cn; bwang@ustc.edu.cn

desired kagome materials[5] has stimulated the development of on-surface chemistry and supramolecular self-assembly for the fabrication of kagome lattices in single layers of metal-organic frameworks (MOFs)[24,25], hydrogen-bond organic frameworks (HOFs)[26–30], and covalent organic frameworks (COFs)[31,32]. While those characterized organic frameworks were imaged to show the structures of kagome lattices, in only a few of them, the electronic properties were observed with FB features but were far from the Fermi level ($E_F$) by approximately 2 eV[30–32]. For practical applications, the fabrication of metallic kagome materials with kagome bands near $E_F$ is highly desired and is also a theoretical concern[33].

Alternatively, 2D artificial electronic lattices, formed with structured surface potentials by precisely patterning nonmetal atoms or molecules that function as repulsive potentials to govern hopping of the confined Shockley surface-state electrons on coinage metals[34], can deterministically deliver designer low-energy electronic and topological properties near $E_F$[1–3]. For example, by the tip-assisted manipulation of carbon monoxide molecules using a scanning tunneling microscope (STM), graphene lattices[35,36], Sierpiński triangles[37], Lieb[38,39] and kagome lattices[40] were constructed, with customized electronic structures and topology. By tackling more sophisticated supramolecular self-assembly protocols[41,42], the bottom-up construction procedure enables the scale-up ability and high tunability of the quantum phases due to mesoscopic templating for quantum confinement and quasiparticle scattering at well-defined interfaces, which has produced many quantum-dot superlattices[43–45]. By further combining the well-developed on-surface synthetic strategy, an electronic chiral kagome-honeycomb lattice has recently been observed in the hexagonal circumcoronene superlattice with high yield and atomic precision[46]. However, there is still a lack of a general strategy to realize varieties of electronic kagome lattices with high geometric and topological tunability in supramolecular framework-patterned metal surfaces.

Herein, we demonstrate a more general on-surface route to constructing highly tunable artificial electronic kagome lattices on coinage metal (111) surfaces (Fig. 1) by self-assembling (quasi)hexagonal organic molecules or bottom-up synthesized carbon-based nanostructures through halogen-coordinated hydrogen bonds, designed as halogen hydrogen-bonded organic frameworks[47] (XHOFs). As a proof of concept, we fabricate three different XHOFs on Ag(111) and Au(111) surfaces, of which the halogen–hydrogen bonding geometries are characterized using STM and bond-resolved noncontact atomic force microscopy (nc-AFM), complemented by density functional theory (DFT) calculations. Emerging electronic kagome lattices, from regular to breathing and chiral breathing diatomic-kagome lattices, are unveiled by scanning tunneling spectroscopy (STS) measurements, showing distinct FB energies in the range of 0.50–1.50 eV. The experimentally observed charge ordering and topological edge states, consistent with the DFT and cluster-model tight-binding (TB) calculations, confirm the feasibility of our design strategy for simulating noteworthy quantum phenomena.

## Results

### Design strategy and on-surface synthesis of XHOFs

Our strategy for designing artificial electronic kagome lattices uses (quasi)hexagonal organic building blocks that self-assemble into XHOFs through halogen-based hydrogen bonds (Fig. 1a). This is accomplished by the templating effect of the XHOFs on coinage metals[1–3], which can regulate the movement of the confined Shockley surface-state electrons within the triangular empty regions surrounded by halogen atoms and organic building blocks, as marked by blue shadowing in Fig. 1a, to satisfy the conditions of emerging artificial kagome lattices. The phase cancellation in the kagome lattices renders the kagome FBs and real-space localization, which reshapes the dispersive band of the Shockley surface state (SS) in the coinage metal surface to diverse kagome bands (Fig. 1b).

Different from MOFs and HOFs, XHOF represents another kind of framework materials that has recently been synthesized in the three-dimensional form from solution-based reactions[47]. Halogen atoms in

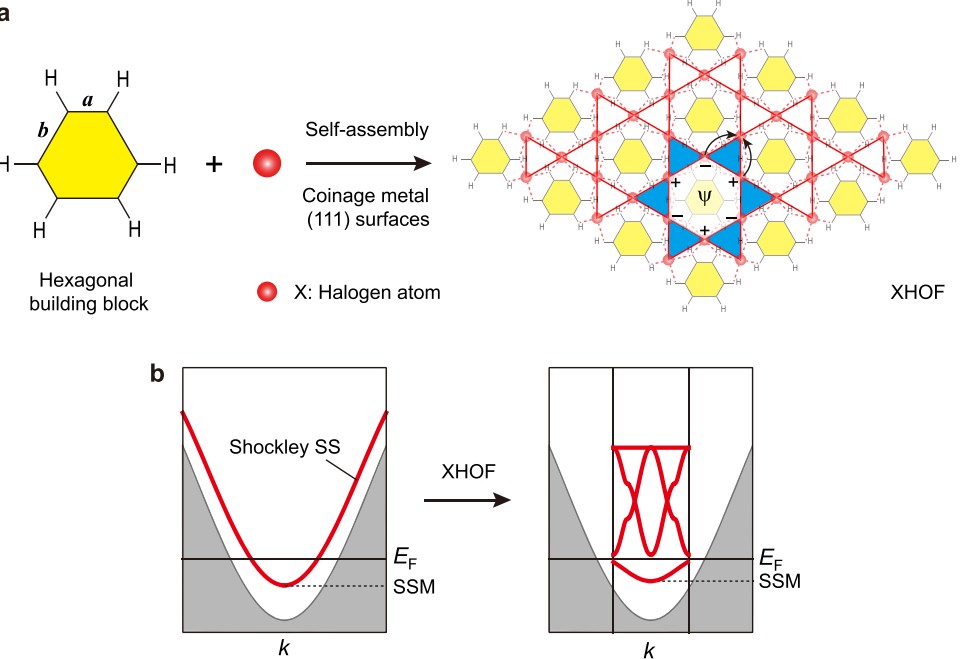

**Fig. 1 | Schematic illustration of the on-surface strategy to construct artificial electronic kagome lattices. a** Hexagonal building blocks, with side lengths $a$ and $b$, self-assemble into halogen-hydrogen bonded organic frameworks (XHOFs) on coinage metal (111) surfaces. The quantum destructive interference, as illustrated by arrows, gives rise to flat bands, with real-space localization within the blue-shadowed triangles. **b** Schematic drawing of the formation of artificial electronic kagome bands by reshaping the dispersive Shockley surface state (SS) from the patterning effect of the XHOF. The dispersive Shockley SS exhibits its minimum (SSM) below the Fermi energy ($E_F$), which is at around −100 meV for the Ag(111) surface and −450 meV for the Au(111) surface, respectively.

XHOFs work as connection nodes, similar to metal atoms in MOFs but different from HOFs, where hydrogen bonds are directly formed between organic molecules. Recent progress in on-surface synthesis provides enormous possibilities for realizing the long-range order of this type of 2D XHOFs based on versatile atomically precise nano-carbon architectures and commonly existing halogen atoms[48]. To achieve the sixfold symmetry of the XHOFs, the (111) surfaces of coinage metals are adopted. Taking advantage of the flexibility of the halogen–hydrogen bonds and tunability of the edge lengths ($a$ and $b$) and the periphery geometries of the organic building blocks, one can easily tailor the absolute and relative values of hopping parameters, changing the electronic regular kagome lattice ($a = b$) to the breathing kagome lattice ($a \neq b$), and even more complicated varieties, with highly tunable FB energies.

The building blocks can either be small hexagonal organic molecules, such as benzene, or larger carbon-based nanostructures synthesized by on-surface reactions, which can simultaneously provide the needed halogen atoms in the XHOFs if halogenated precursor molecules are used. As a proof of concept, we first use benzene molecules as the building block (Fig. 2a), which are deposited onto a Ag(111) surface with preexisting Br atoms (see Methods). The STM (Fig. 2b and Supplementary Fig. 1) and nc-AFM images (Fig. 2c, d and Supplementary Fig. 2) show that each benzene molecule is surrounded

by six Br atoms and stabilized by twelve Br···H bonds in the supra-molecular assembly, namely, the benzene/Br/Ag(111) superlattice, which is well reproduced by the DFT-simulated structural model (Fig. 2e). By utilizing the well-developed on-surface synthesis approach, we can atomically precisely grow various building blocks that can effectively bond to the simultaneously generated halogen atoms and self-assemble into XHOF superlattices. Consequential annealing of 5,8-dibromopicene (DBP) molecules at 473 and 573 K on Ag(111) and Au(111) surfaces (Fig. 2f, k), respectively, results in quasi-hexagonal organometallic trimers M-C66 (Fig. 2g and Supplementary Fig. 1) and porous nanographenes C66 (Fig. 2l and Supplementary Fig. 1). The bond-resolved nc-AFM images clearly reveal the chemical bonding geometries of M-C66 (Fig. 2h, i and Supplementary Fig. 2) and C66 (Fig. 2m, n and Supplementary Fig. 2) and the Br···H bond networks, as captured by the simulated structural models of the M-C66/Br/Ag(111) (Fig. 2j) and C66/Br/Au(111) superlattices (Fig. 2o). Considering the relatively large electronegativity of the Br atom, its brightness in the nc-AFM images is quite low, suggesting charge transfer with the surrounding molecules.

The hexagonal XHOF superlattices (Fig. 2b, g, l) can effectively pattern the surface potential landscapes with the formation of trian-gular regions of the exposed metal surfaces, as depicted in Fig. 2e, j, o by yellow/blue shadowing, which are defined by the surrounding Br

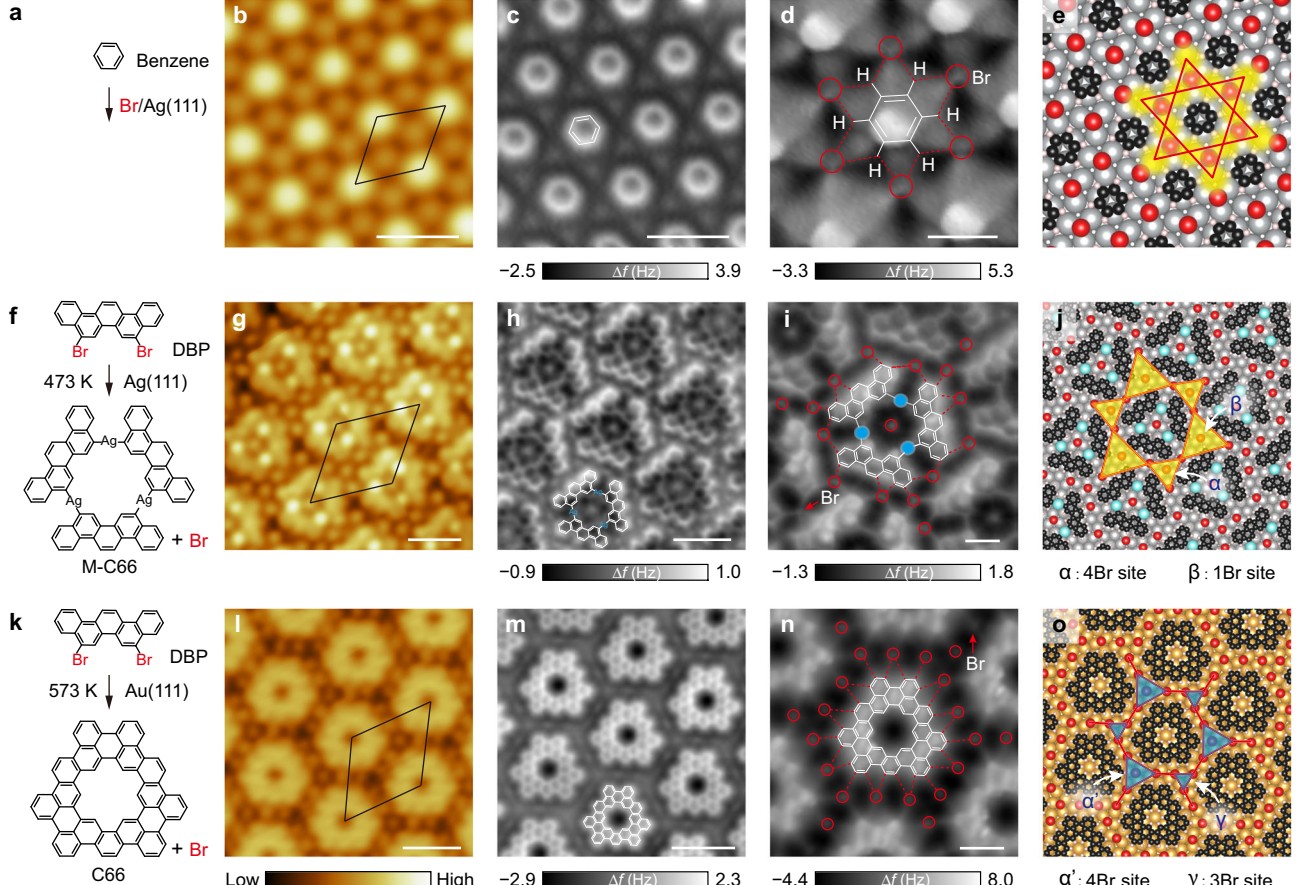

**Fig. 2 | Construction of XHOF superlattices. a**, **f**, **k** Schematics of the on-surface synthesis, **b**, **g**, **l** STM images, **c**, **h**, **m** nc-AFM images, **d**, **i**, **n** zoom-in nc-AFM images with closer tips, and **e**, **j**, **o** DFT-relaxed structural models of the benzene/Br/Ag(111) superlattice (**a–e**), the M-C66/Br/Ag(111) superlattice (**f–j**), and the C66/Br/Au(111) superlattice (**k–o**). STM imaging conditions: **b** $V_s = 100$ mV, $I_t = 200$ pA; **g** $V_s = -100$ mV, $I_t = 1$ nA; **l** $V_s = -800$ mV, $I_t = 10$ pA. All nc-AFM images were acquired with a CO-functionalized tip, with tip heights $\Delta z = -10$ (**c**), $-20$ (**d**), 30 (**h**), $-5$ (**i**), $-20$ (**m**), and $-30$ pm (**n**), respectively, with respect to the setpoint condition

$V_s = -0.8$ V, $I_t = 10$ pA on molecules. In **d**, **i**, and **n**, the Br···H bonds and the positions of Br atoms highlighted by the red dashed lines and the red circles/arrows, with the raw images shown in Supplementary Fig. 2. In **e**, **j**, and **o**, the yellow and blue shadowing indicate confined sites for surface electrons that form corresponding kagome lattices, and the atoms are color-coded: gray or light cyan for Ag, dark for C, pink for H, red for bromine, and gold for Au. Scale bars: **b**, **c** 1 nm; **d** 0.5 nm; **g**, **h** 2 nm; **i** 1 nm; **l**, **m** 2 nm; **n** 1 nm.

atoms and the molecular structures in the presence of Br⋯H bond networks. Following our design strategy, these XHOF superlattices should give rise to different varieties of electronic kagome lattices. For the case of the benzene/Br/Ag(111) superlattice, the sixfold symmetry indicates a regular kagome lattice (Fig. 2e). In contrast, for the M-C66/Br/Ag(111) superlattice, the threefold symmetry generates the breathing kagome lattice (Fig. 2j), where the corner-sharing triangles have different sizes and Br occupations, namely, the 4Br (α) and 1Br sites (β), with a small tilting angle of approximately 7°. For the C66/Br/Au(111) superlattice, the chiral breathing diatomic-kagome lattice can emerge (Fig. 2o), with different triangle sizes, i.e., the 3Br (γ) and 4Br sites (α′), and the chirality induced by the tilting angle of approximately 30° between the triangles.

### Theoretical insight into the electronic kagome lattices

To further shed light on our strategy for the electronic kagome lattices, we simulate the electrostatic potentials of the three superlattices (Fig. 3a, c, e) using the DFT method based on the structural models obtained in Fig. 2. The regions occupied by Br atoms or molecules show negative charge accumulation, which is balanced by positive charge accumulation in the exposed metal surfaces, suggesting consistent electron transfer from substrates to adsorbents. Electronic lattices formed by the confined 2D electron gases due to the presence of XHOFs display the characteristic patterns of the kagome lattice (Fig. 3a), the breathing kagome lattice (Fig. 3c), and the chiral breathing diatomic-kagome lattice (Fig. 3e).

Considering our DFT calculations that include semi-infinite metal surfaces with only four metal-atom layers, due to a computational feasibility, cannot accurately describe the electronic structures of the Shockley SSs[49] and the confined states[46] in the XHOF superlattices, we calculate the band structures of the three electronic kagome lattices by adopting the TB theory that involves the electron hopping strength between atomic orbitals associated to lattice sites. The kagome lattice with hopping constants $t_1 = t_2 = 0.175$ eV exhibits the characteristic features of an FB through the Brillouin zone, van Hove singularities at

the M point, and the Dirac point at the K point, with energies of approximately 1.05, 0.36/0.71, and 0.53 eV, respectively (Fig. 3b). Introduction of different hopping constants with $t_1 = 0.175$ eV and $t_2 = 0.20$ eV opens a gap at the Dirac point with a size of approximately 0.08 eV (Fig. 3d) in the breathing kagome lattice. For the chiral breathing diatomic-kagome lattice, an additional constant is required to describe the hopping between the triangles. By choosing $t_1 = t_2 = 0.50$ eV and $t_3 = 0.60$ eV, we obtain two FBs located at zero energy (FB1) and 1.00 eV (FB2) and a gap at the Dirac point, showing the characteristic four-band structure (Fig. 3f). The band structures are independent on the tilting angles between triangles observed in the experiment (Fig. 3d, f).

### Regular and breathing kagome superlattices

The real-space localization of the electronic kagome lattices can be probed by STS measurements. The d$I$/d$V$ spectra in Fig. 4a acquired in the benzene/Br/Ag(111) superlattice suggest a significant enhancement of the local density of states (LDOS) at an energy of 1.46 eV, which show comparable intensities measured at exposed surface between Br atoms or benzene molecules (magenta) and on the Br atom (green), but slightly weaker on the benzene molecule (blue) (Supplementary Fig. 3). As a reference, the Shockley SS on the Ag(111) surface displays a characteristic step-like feature in the d$I$/d$V$ spectrum with the minimum (SSM) at around −80 meV. According to the DFT simulations (Supplementary Fig. 4), the bandgap of the benzene molecule should be around 5 eV, with the highest occupied molecular orbital (HOMO) and the lowest unoccupied molecular orbital (LUMO) at −2.29 and 2.67 eV, respectively, which suggests that the observed intense peak at 1.46 eV should not originate from the orbitals of the benzene molecule. The d$I$/d$V$ map acquired at 1.35 eV, corresponding to the rising edge of the observed peak to increase the contrast between different sites[50] (Supplementary Fig. 3), displays the kagome lattice with enhanced LDOSs at the exposed Ag surface (Fig. 4c and Supplementary Fig. 5), which is consistent with the simulated electrostatic potential (Fig. 3a). This observation indicates that the observed state results from the

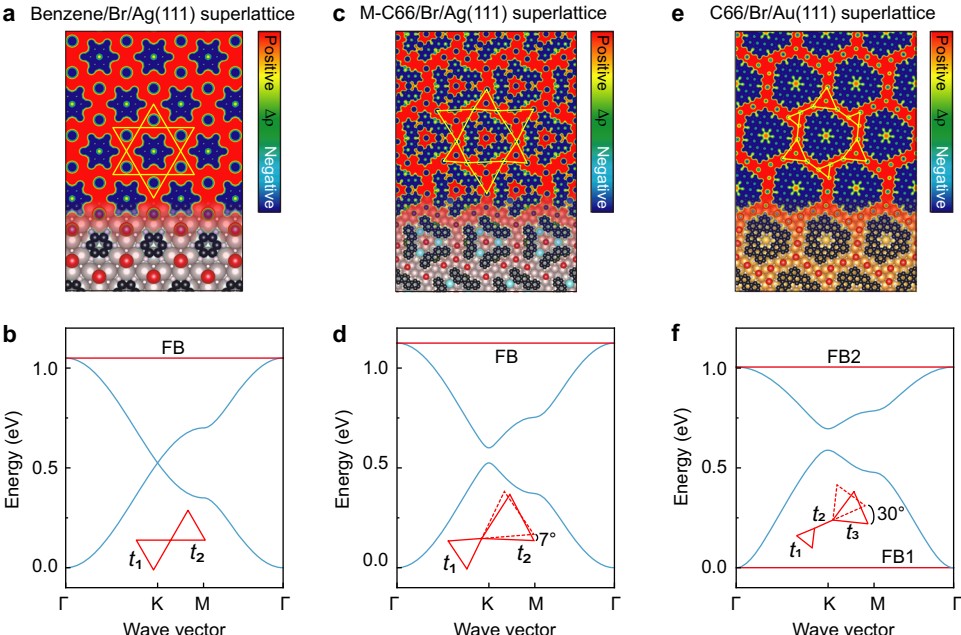

**Fig. 3 | Theoretical calculations.** DFT-simulated electrostatic potentials, Δρ, of the benzene/Br/Ag(111) superlattice (**a**), the M-C66/Br/Ag(111) superlattice (**c**), and the C66/Br/Au(111) superlattice (**e**), respectively, superimposed with the structural model at bottom of each panel. The $k$-space dispersion of the electronic bands of the kagome lattice (**b**), the breathing kagome lattice (**d**), and the chiral breathing

diatomic-kagome lattice (**f**), respectively, calculated using the TB model with different tilting angles and hopping constants $t_1 = t_2 = 0.175$ eV (**b**), $t_1 = 0.175$ eV, $t_2 = 0.20$ eV (**d**), and $t_1 = t_2 = 0.50$ eV, $t_3 = 0.60$ eV (**f**). The flat bands (FBs) are highlighted with red.

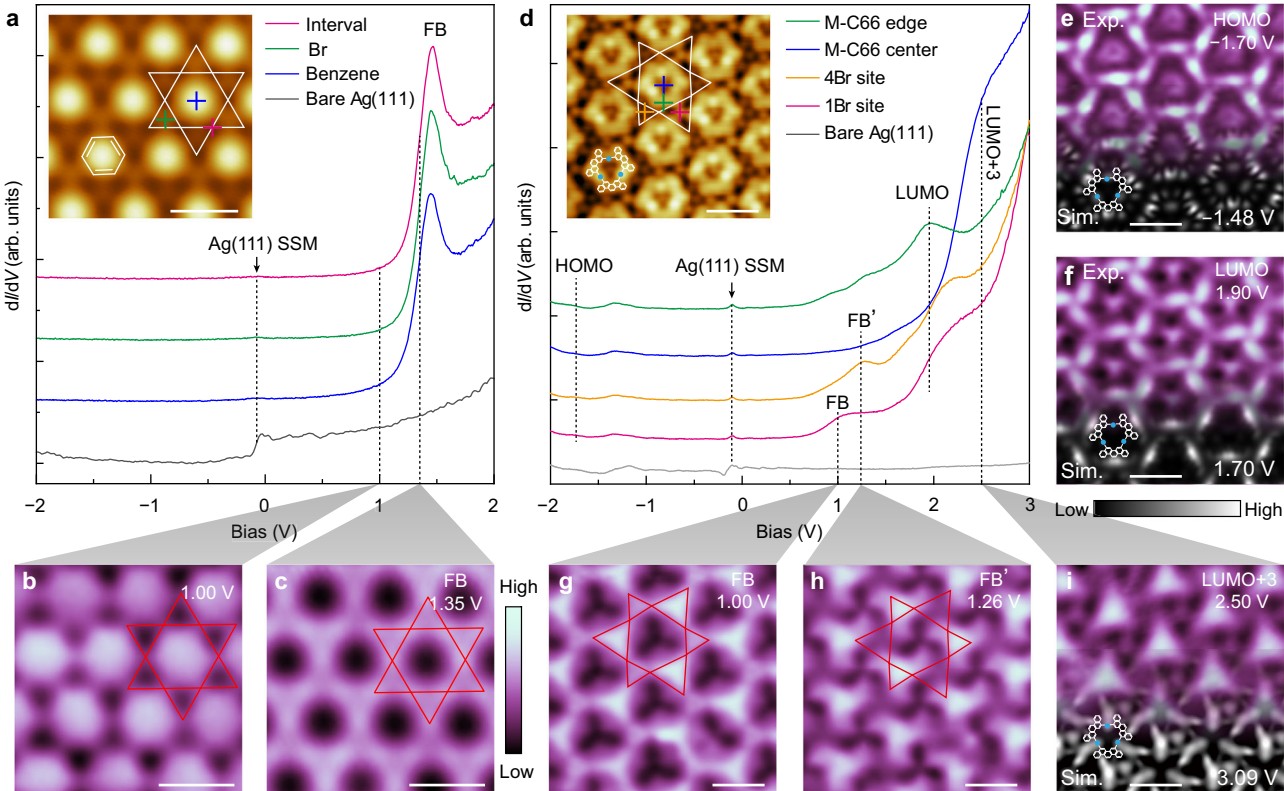

**Fig. 4 | Electronic characterizations of the benzene/Br/Ag(111) and M-C66/Br/Ag(111) superlattices.** d$I$/d$V$ spectra of the benzene/Br/Ag(111) superlattice (**a**) and the M-C66/Br/Ag(111) superlattice (**d**), acquired at colored-cross marked positions in the corresponding inset STM images, with gray curves from the bare Ag(111) surface as a reference. In **a** and **d**, the minimum of the dispersive Shockley surface state (SSM) of the Ag(111) surface around −80 meV is marked. The spectra are shifted vertically for clarity. d$I$/d$V$ measurement parameters: **a** $V_s$ = 2.00 V, $I_t$ = 50 pA; **d** $V_s$ = −2.50 V, $I_t$ = 10 pA. STM imaging conditions: **a** $V_s$ = 1.00 V, $I_t$ = 50 pA;

**d** $V_s$ = −130 mV, $I_t$ = 50 pA. **b, c** d$I$/d$V$ maps obtained at energies of 1.00 and 1.35 eV, respectively, as marked in each figure. **e**–**i** d$I$/d$V$ maps obtained at energies of the highest occupied molecular orbital (HOMO) of −1.70 eV, the lowest unoccupied molecular orbital (LUMO) of 1.90 eV, two flat bands of 1.00 (FB) and 1.26 eV (FB'), and the LUMO + 3 orbital of 2.50 eV, as indicated in **d**. In **e, f,** and **i**, simulated LDOS maps corresponding to the HOMO (−1.48 eV), LUMO (1.70 eV) and LUMO + 3 (3.09 eV) of M-C66, respectively, are superimposed at the bottom. Scale bars: **a**–**c** 1 nm; **d**–**i** 2 nm.

confined Shockley SS. In contrast, there is no contribution from the confined Shockley SS (seen by the dim triangles) at a slightly lower energy of 1.00 eV (Fig. 4b). Hence, the pronounced peak centered at 1.46 eV can be reasonably assigned to the kagome flat band from the confined Shockley SS. It is noted that similar levels from the confined Shockley SSs were observed before, but they were simply attributed to their upward shifts[43,45]. We believe that the much large energy shifts of these states in our experiments should reveal the origin because of the strong interaction in the kagome lattice. The observable peak intensity on the benzene molecule results from the penetration of confined electrons due to the small size of the benzene molecule. This may reduce the confinement of the patterned surface states and broaden the emerging kagome FB bandwidth, which can be improved by adopting larger building blocks, as shown in the following. Nevertheless, this proof-of-concept result confirms the feasibility of our design strategy.

The M-C66/Br/Ag(111) superlattice presents richer electronic states in the d$I$/d$V$ spectra (Fig. 4d and Supplementary Fig. 6). The Ag(111) SSM is consistently observed over the superlattice, with slightly different energies and shapes compared to those of the SSM on the pristine surface, suggesting that the modification of the band bottom of the dispersive Shockley SS is small in the presence of the XHOF. On the M-C66, the HOMO and LUMO are found at −1.70 and 1.90 eV, respectively, which are confirmed by the corresponding d$I$/d$V$ maps and simulated LDOS maps (Fig. 4e, f and Supplementary Fig. 7). Note that, due to the relatively weak peak features in the d$I$/d$V$ spectra, the HOMO position of about −1.70 eV is assigned by comparing a serial of

d$I$/d$V$ maps acquired at various negative biases, at which the more pronounced molecular frontier orbitals on the M-C66 show up (Supplementary Fig. 6). At the inequivalent 1Br and 4Br sites, we observe two different peaks at 1.00 and 1.26 eV, respectively, within the molecular HOMO−LUMO gap, suggesting the greatly enhanced effective masses of the confined Shockley surface states at these energies due to the formation of kagome flat bands. Consistently, the d$I$/d$V$ map at 1.00 eV (FB) reveals the breathing kagome lattice (Fig. 4g), as expected from our design model, where the LDOS intensities are inverse to the local Br concentrations. Interestingly, the state at 1.26 eV (FB') displays a clear chiral feature (Fig. 4h), with spatial LDOS distributions almost reversed to those at 1.00 eV. This suggests that these two states probably result from inversion symmetry breaking in the M-C66/Br/Ag(111) superlattice. In addition, one can see that the FBs are significantly suppressed at the central hole of the M-C66 structure, where a strong LDOS enhancement is observed at approximately 2.50 eV (Fig. 4d, i). It can be safely assigned to the forth lowest unoccupied molecular orbital (LUMO + 3) of the M-C66, after compared with the simulated spatial distributions of various molecular orbitals (Supplementary Fig. 7).

## Chiral breathing diatomic-kagome lattice
While possessing the same threefold symmetry as M-C66, the distinct peripheral geometry of C66 changes the halogen−hydrogen bond network that significantly modifies the surface electronic potential landscape, leading to the chiral breathing diatomic-kagome lattice in the C66/Br/Au(111) superlattice. This particular variety of the kagome

lattice presents a distinct feature—compared with that of the regular or breathing kagome lattices—that is, the emergence of two FBs (Fig. 3f), which also offer the signature for the experimental confirmation of this specific kagome lattice. By combining d$I$/d$V$ measurements (Fig. 5a, b) and DFT calculations (Supplementary Fig. 8) of the C66/Br/Au(111) superlattice, we can assign the HOMO at −0.87 eV (Fig. 5c), LUMO at 1.85 eV (Fig. 5d), and LUMO + 2 at 2.57 eV (Fig. 5g), which are also consistent with our previous observations of the isolated C66 nanographenes[51]. Here, the d$I$/d$V$ spectra are normalized by (d$I$/d$V$)/($I$/$V$) to suppress the strong intensity of the LUMO + 2 orbital (Supplementary Fig. 9). In addition, the energy upward shift of Au(111) SSM from −0.45 eV to −0.27 eV in the C66/Br/Au(111) superlattice is also more clearly displayed (Fig. 5a and Supplementary Fig. 9), which is in line with the modified boundary conditions[52–54]. Typically, the HOMO orbital (Fig. 5c) displays the chirality of the three-petal pinwheel-like patterns at the inequivalent 3Br and 4Br sites. The LUMO + 2 orbital (Fig. 5g) is well localized at the central nanopore of C66, which forms a quantum-dot-like superlattice in XHOF, similar to the LUMO + 3 orbital in the M-C66/Br/Ag(111) superlattice (Fig. 4i). In addition to the molecular orbitals, we observe two peaks within the HOMO−LUMO gap at approximately 0.50 and 1.20 eV. Figure 5e, f shows the d$I$/d$V$ maps

acquired at 0.50 (FB1) and 1.20 eV (FB2), respectively, displaying the same chiral breathing diatomic-kagome lattice. The observed distinct FB1 and FB2 states (Supplementary Fig. 10) agree well with the two predicted FBs in the TB band structure (Fig. 3f). Notably, the 1.20 eV state (FB2) shows a small energy variation of approximately 0.10 eV between the 3Br and 4Br sites (Fig. 5a) when compared with that of 0.26 eV between the 1Br and 4Br sites in the M-C66/Br/Ag(111) super-lattice (Fig. 4d), suggesting that the local Br concentration difference can effectively tune FB energies and break the inversion symmetry in the artificial kagome lattices.

The emergence of electronic kagome lattices relies on two recipes. One is the potential barriers induced by the molecular nanostructures, which means that the emerging kagome bands should reside in the HOMO−LUMO gaps, in line with the experimental observations (Figs. 3–5). The other is the charge transfer from metal substrates to molecular structures in the XHOFs, which consistently induces charge ordering, as exhibited by the calculated electrostatic potentials (Fig. 3a, c, e). We further verify this critical phenomenon in the C66/Br/Au(111) superlattice. The d$I$/d$V$ spectra collected on C66 and at the interval between C66 within a small energy window (±0.25 eV) show reversed LDOS intensities for the two sites below and

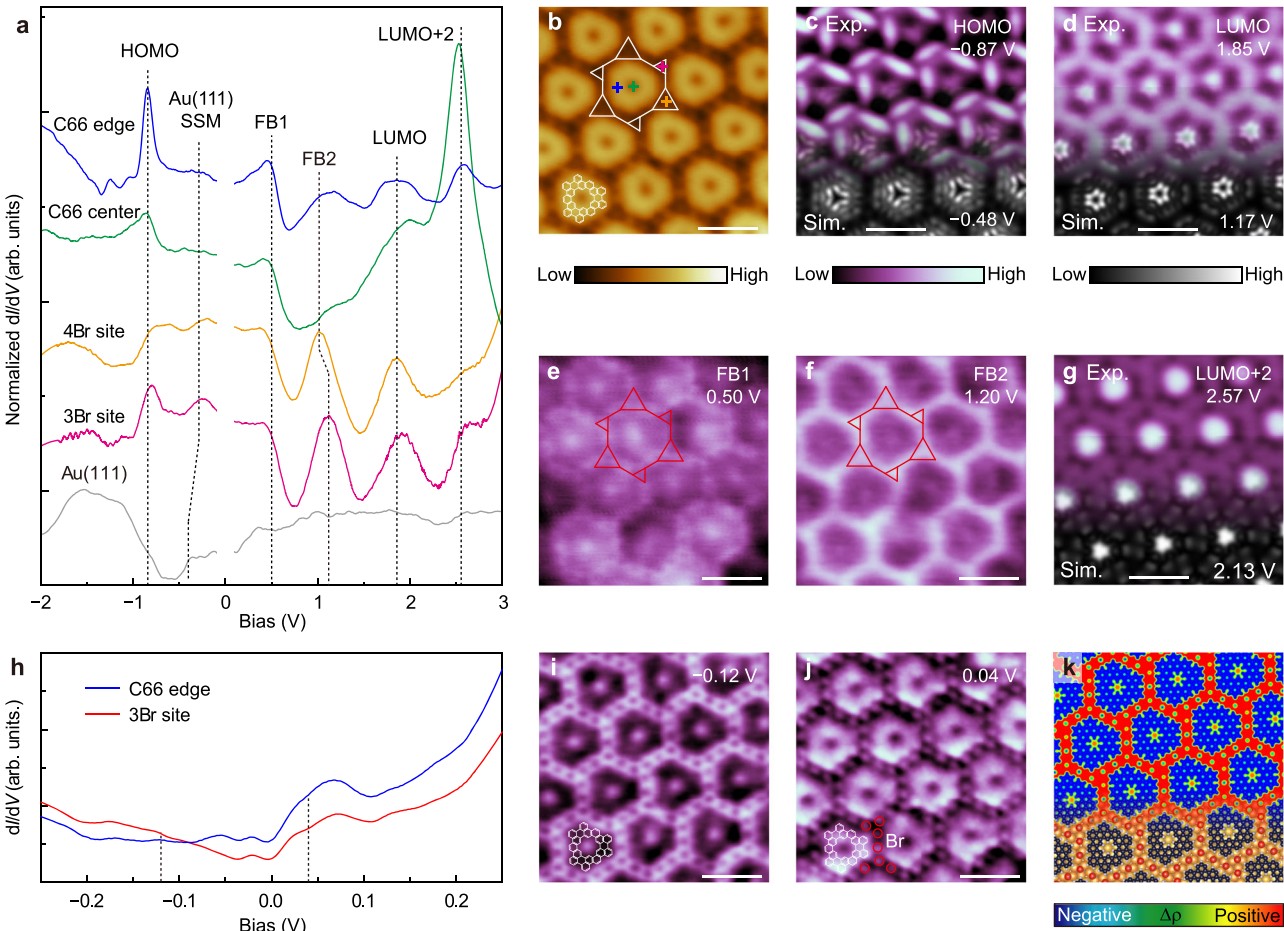

**Fig. 5 | Electronic characterizations of the C66/Br/Au(111) superlattice.**
**a** Normalized (d$I$/d$V$)/($I$/$V$) spectra, **b** high-resolution STM image, and **c**–**g** d$I$/d$V$ maps obtained at energies of −0.87 (HOMO), 1.85 (LUMO), 0.50 (FB1), 1.20 (FB2), and 2.57 eV (LUMO + 2), respectively, as indicated in **a**. d$I$/d$V$ measurement parameters: **a** $V_s$ = −2.00 V, $I_t$ = 10 pA; **c**–**g** $I_t$ = 10 pA. STM imaging conditions: **b** $V_s$ = −120 mV, $I_t$ = 10 pA. In **a**, the narrow region near $E_F$ (zero bias) is not shown, due to some uncertain features induced during the normalization. Another set of data recorded using a different tip is also shown in Supplementary Fig. 10 for comparison, which gives the tip-independent features in the spectra and maps that

are in good agreement with those in **a**–**g**. In **c**, **d**, and **g**, simulated LDOS maps corresponding to the HOMO (−0.48 eV), LUMO (1.17 eV), and LUMO + 2 (2.13 eV) of the C66 nanographene, respectively, are superimposed at the bottom. **h** d$I$/d$V$ spectra with a narrow energy window acquired from the C66/Br/Au(111) super-lattice. d$I$/d$V$ measurement parameters: **h** $V_s$ = −2.00 V, $I_t$ = 10 pA; **i**, **j** $I_t$ = 10 pA. **i**, **j** d$I$/d$V$ maps obtained at energies of −0.12 and 0.04 eV, respectively, as indicated in **h**. **k** Simulated electrostatic potential of the C66/Br/Au(111) superlattice corresponding to the same area as **i**, **j** with the structural model superimposed at the bottom. Scale bars: 2 nm.

above an energy of approximately −0.09 eV, very close to $E_F$ (Fig. 5h). Correspondingly, the d$I$/d$V$ maps taken at energies of −0.12 (Fig. 5i) and 0.04 eV (Fig. 5j) display the dominant LDOS intensities in the interval regions between C66 and on C66, respectively. Thus, there is an apparent out-of-phase feature between the occupied (Fig. 5i) and unoccupied states (Fig. 5j) in real space, which is a signature of charge order[55]. Their distribution patterns correspondingly resemble the simulated positive and negative charge accumulation regions (Fig. 5k), directly confirming our design strategy for the artificial kagome lattices. A similar observation of the charge order in the M-C66/Br/Ag(111) superlattice is presented in Supplementary Fig. 11. The presence of charge orders in the XHOFs may be suggestive of the correlation effect near $E_F$.

## Edge states in the electronic kagome lattices

Gap opening in the electronic breathing kagome lattices suggests that topologically protected edge or corner states may emerge in the gaps[15,18,40], different from the case of the semimetallic kagome lattice. Figure 6a, e, i and Supplementary Fig. 12 show that all three XHOF islands present enhanced LDOSs at the edges within the energy ranges of the Dirac points or the gaps (Fig. 3). However, one should keep in mind that there may be trivial states at the periphery of the XHOF

islands, while the topological edge states are located at the kagome lattice edges[14]. By d$I$/d$V$ mapping at energies of the edge states and FBs, we can distinguish spatial distributions of the edge states and the kagome lattices, which helps to unveil the origins of the observed edge states. Figure 6b, d shows d$I$/d$V$ maps taken at 0.80 and 1.50 eV, respectively, in the benzene/Br/Ag(111) superlattice, which suggest that the edge state is mainly contributed by the trivial states located at the XHOF periphery, such as the hybridization states between edge Br atoms and the Ag(111) surface. In line with this assignment, the TB cluster model calculation shows relatively weak and uniform edge states around the kagome lattice (Fig. 6c), which is similar to the dangling-bond states[16].

For the M-C66/Br/Ag(111) superlattice, there are two different terminations of the breathing kagome lattice edges (Fig. 6f, h). One is terminated with the bright large triangles, and the other is terminated with the weak small triangles, while only the latter edge hosts the topological in-gap states (Fig. 6g). Note that the bright triangles with enhanced LDOSs in experiment suggest larger hopping $t$, corresponding to the smaller triangles in the TB models. After carefully comparing the two different edges in the experimental d$I$/d$V$ map (Fig. 6f), we indeed observe additional LDOSs at the edges with weak small triangles (solid red rectangles) compared to those with bright

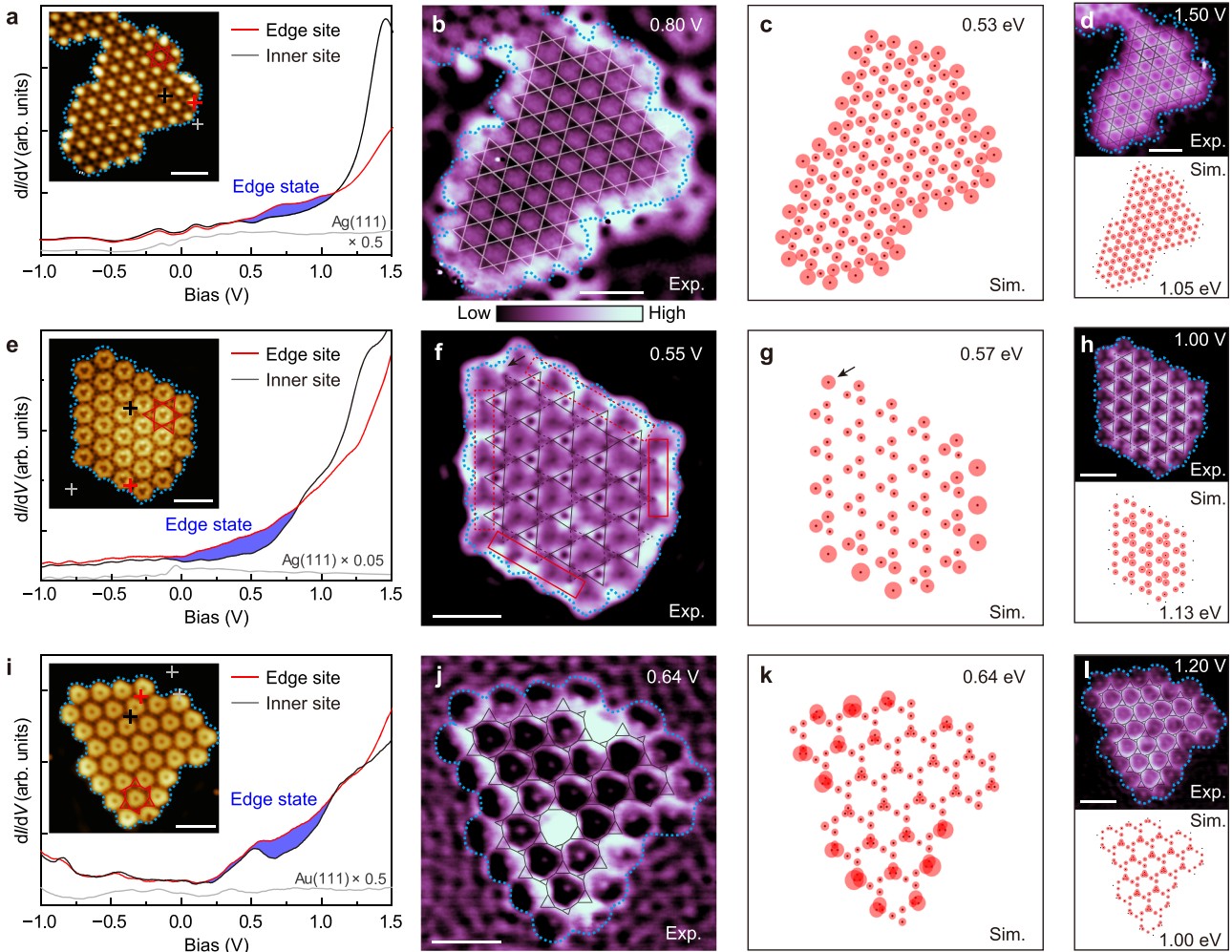

**Fig. 6 | Edge states in the XHOF islands. a, e, i** Experimental d$I$/d$V$ spectra, **b, f, j** d$I$/d$V$ maps taken at energies of the corresponding edge states, **c, g, k** TB-simulated LDOS maps of the edge states, **d, h, l** experimental d$I$/d$V$ (top) and simulated LDOS (bottom) maps taken at energies of the corresponding bulk states for the benzene/Br/Ag(111) superlattice (**a**–**d**), the M-C66/Br/Ag(111) superlattice (**e**–**h**), and the C66/Br/Au(111) superlattice (**i**–**l**), respectively. In **a, e, i**, the blue shadowing marks the

edge states. The blue dotted lines in experimental images indicate the periphery of XHOF islands. The complete electronic kagome lattices are also superimposed on the corresponding maps. STM imaging conditions: **a** $V_s$ = 1.50 V, $I_t$ = 500 pA; **e** $V_s$ = −0.13 V, $I_t$ = 50 pA; **i** $V_s$ = −0.12 V, $I_t$ = 10 pA. d$I$/d$V$ measurement parameters: **a** $V_s$ = −2.00 V, $I_t$ = 700 pA; **e** $V_s$ = 2.00 V, $I_t$ = 100 pA; **i** $V_s$ = −2.00 V, $I_t$ = 10 pA. Scale bars: **a, b, d** 2 nm; **e, f, h** 4 nm; **i, j, l** 4 nm.

large triangles (dashed red rectangles). Notably, at the top-left corner, both the experimental and simulated results show enhanced LDOS (black arrows in Fig. 6f, g), which may be assigned to the high-order topological corner states in the breathing kagome lattice[17,18,40]. While the presence of hybridization states scattered around the peripheries of the superlattice islands (blue dotted lines in Fig. 6b, f) may disturb the above assignments, the absence of hybridization states at the C66/Br/Au(111) superlattice periphery provides undoubted evidence of the presence of topological edge states in the chiral breathing diatomic-kagome lattice (Fig. 6j–l). In line with these assignments and previous theoretical predictions[15,18,40], our TB calculations confirm the topological edge states emerging in the gap regions of the breathing kagome lattices (Supplementary Fig. 13). These results substantiate that the artificial lattices emerging from the XHOFs on metal surfaces can effectively simulate the electronic and topological properties of complex quantum materials.

## Discussion

The physical mechanism underlying our strategy for creating artificial electronic kagome lattices is similar to the artificial electronic lattices patterned by CO molecules on metal surface via STM tip-assisted manipulation[35–40]. Notably, while the STM manipulation provides an ultimate control over the lattice topology, it has obvious drawbacks, including being time-consuming, laborious, inefficient, and lacking scalability. However, our approach could surpass those drawbacks to generate variety of electronic kagome lattices with huge complexity, such as the chiral breathing diatomic-kagome lattice that was not realized before, and to achieve large domain sizes of the kagome lattices for up to 100 nm or more (Supplementary Fig. 1). Another advantage of our strategy lies in the fact that our strategy can be further seamlessly combined with the well-developed on-surface chemistry. This offers the almost limitless tunability for the symmetries and peripheral shapes of the organic building blocks in the XHOFs, which enables tremendous varieties of the artificial kagome lattices. These exciting aspects offer great opportunities for simulating and exploring the electronic and topological properties of complex quantum materials. The 2D XHOFs also open a door to assemble the atomically precise on-surface synthesized nanostructures for realizing not only collective properties but also practical multifunctionalities[56–58], considering the porous structures that can be periodically embedded with individual or trimerized metal atoms and other functional groups[59]. We also need to admit that both our strategy and the tip-assisted manipulation approaches suffer from the broadening effect of the confined electronic states and the possibly emerging kagome flat bands due to the surface-bulk scattering of electrons that results in the limited coherent lifetime on the metallic surface[3,60], as observed in our experiment with finite peak widths ranging from about 0.2–0.4 eV and in previous reports with similar values[40,46].

In summary, we have demonstrated a general strategy to realize artificial electronic kagome lattices with diverse varieties, by utilizing versatile XHOFs to pattern surface electronic potential landscapes on coinage metal (111) substrates. Three proof-of-concept XHOF superlattices and the corresponding distinct electronic kagome lattices are constructed, from the regular to breathing kagome lattices and the chiral breathing diatomic-kagome lattice, with emerging topological edge states and charge orders, which substantiate the reliability of our design strategy. We envision that this strategy that combines on-surface chemistry and in situ scanning probe microscopy can be used to achieve various artificial electronic lattices, such as the Lieb lattice[38,39] and fractal lattices[37], by varying the symmetries of the building blocks and metal substrates. Therefore, our work provides a general on-surface route to fabricating artificial lattices with designer electronic and topological properties and with obvious advantages, such as scale-up ability and atomic precision, which offer great

opportunities for exploring topological FB physics in complex quantum phases of matter.

## Methods

### Sample preparation

The on-surface synthesis was performed under ultrahigh-vacuum (UHV) conditions with a base pressure better than $5 \times 10^{-11}$ mbar. The Ag(111) and Au(111) substrates (MaTeck GmbH) were cleaned by repeated cycles of Ar$^+$ sputtering (1.5 kV) and annealing (750 K). The liquid benzene molecules were deposited onto the brominated Ag(111) surface held at 5 K through a leak valve into the UHV chamber at a pressure of approximately $1 \times 10^{-10}$ mbar for 10 s. The brominated Ag(111) surface was prepared through Ullmann-like reactions of multiple Br-containing molecules on clean Ag(111) at approximately 423-473 K. The DBP molecules in a quartz crucible were thermally evaporated at 394 K onto clean Ag(111) and Au(111) substrates held at room temperature[51]. The typical evaporation time was 1.5-3 min at a pressure of approximately $1.5 \times 10^{-10}$ mbar. Then, the samples were sequentially annealed at 473 and 573 K for approximately 30 min on Ag(111) and Au(111), respectively, to form triangular M-C66 and C66.

### STM/STS and qPlus nc-AFM experiments

STM images and d$I$/d$V$ maps were acquired with a low-temperature scanning tunneling microscope (Scienta Omicron) operated at 5 K in the constant-current mode using a chemically etched and well-cleaned polycrystalline tungsten tip. Sample bias voltages were used with respect to the tip. All the d$I$/d$V$ spectra and maps were obtained using a lock-in amplifier with a modulation of 7 mV at a frequency of 731 Hz. The qPlus AFM measurements were performed with a tungsten tip placed on a qPlus tuning fork sensor[61]. The sensor was driven at its resonance frequency (27,815 Hz) with a constant amplitude of 60 pm. The tip was functionalized with a single CO molecule at the tip apex, picked up from the CO predosed surface. The frequency shift $\Delta f$ from the resonance of the tuning fork was recorded in constant-height mode using Specs Nanonis electronics.

### Density functional theory calculations

The calculations were performed using the Vienna ab initio simulation package[62,63] (VASP) within the framework of density functional theory (DFT). Plane waves with a cut-off energy of 500 eV were used as the basis functions to solve the Kohn-Sham equations. The projector augmented wave (PAW) method[64] was employed to describe the interactions between ionic cores and valence electrons. The exchange-correlation effects were described by the Perdew−Burke−Ernzerhof (PBE) functional[65], and the zero damping DFT-D3 method of Grimme et al.[66] was employed to describe van der Waals interactions. The lattice parameters of bulk Ag and Au were calculated to be 4.140 and 4.155 Å, respectively, which were in good agreement with experiments and previous calculations[67]. The slab models used to simulate the Ag(111) and Au(111) substrates contained four atomic layers plus a sufficiently thick vacuum space (12 Å for atomic relaxation and 20 Å for electronic structure analysis). During atomic relaxation, the bottom two layers were kept frozen, while the coordinates of all other atoms, including those of the adsorbates, were relaxed until the maximum force was less than 0.03 eV Å$^{-1}$. In the simulation of the Ag(111) substrate for the adsorption of the M-C66/Br/Ag(111) superlattice, each atomic layer contained 57 Ag atoms after rotating the supercell by 6.78 degrees, and the Brillouin zone (BZ) was sampled using $2 \times 2 \times 1$ Monkhorst-Pack grids[68]. In the simulation of the Au(111) substrate for the adsorption of the C66/Br/Au(111) superlattice, a $7 \times 7$ supercell was used, and BZ sampling was carried out by using $2 \times 2 \times 1$ Monkhorst−Pack grids. In the simulation of Ag(111) for the adsorption of the benzene/Br/Ag(111) superlattice, a $3 \times 3$ supercell was used, and BZ sampling was performed using $5 \times 5 \times 1$ Monkhorst−Pack grids.

## Tight-binding calculations

We performed tight-binding (TB) calculations on clusters of the kagome lattice, breathing kagome lattice and chiral breathing diatomic-kagome lattice, as shown in Supplementary Fig. 13, where the kagome lattice is a special case of the breathing lattice with uniform hopping amplitudes. The TB Hamiltonian on the kagome lattice is[69]

$$H = - t_1 \sum_{\langle i,j \rangle} c_i^\dagger c_j \tag{1}$$

On the breathing kagome lattice, the Hamiltonian is written as[17]

$$H = -t_1 \sum_{\langle i,j \rangle}^{\text{intra-cell}} c_i^\dagger c_j - t_2 \sum_{\langle i,j \rangle}^{\text{inter-cell}} c_i^\dagger c_j \tag{2}$$

while on the chiral breathing diatomic-kagome lattice, the Hamiltonian reads

$$H = - t_1 \sum_{\langle i,j \rangle}^{\text{large-triangle}} c_i^\dagger c_j - t_2 \sum_{\langle i,j \rangle}^{\text{inter-triangle}} c_i^\dagger c_j - t_3 \sum_{\langle i,j \rangle}^{\text{small-triangle}} c_i^\dagger c_j \tag{3}$$

where $c_i^\dagger$ ($c_i$) is the creation (annihilation) operator at lattice site $i$. Parameters $t_1, t_2, t_3$ represent the hopping amplitudes (Supplementary Fig. 13). In the kagome lattice model, we set $t_1 = 0.175$ eV. In the breathing kagome lattice model, we set $t_1 = 0.175$ eV and $t_2 = 0.2$ eV. In the chiral breathing diatomic-kagome lattice model, we set $t_1 = t_2 = 0.5$ eV and $t_3 = 0.6$ eV.

To obtain the band structure of the kagome lattice, it is necessary to consider the periodic structure of the system. In the following notation, we use Latin letters to denote atomic orbitals in the primitive cell, while (,) denotes the Cartesian coordinates of 2D Bravais lattice points.

The Bloch eigenstate $\psi_{n\mathbf{k}}$ with band index $n$ at momentum $\mathbf{k}$ is given by:

$$\psi_{n\mathbf{k}}(\mathbf{r}) = \sum_\alpha C_\alpha^{n\mathbf{k}} \chi_\alpha^{\mathbf{k}}(\mathbf{r}) \tag{4}$$

where $\chi_\alpha^{\mathbf{k}}$ is the Bloch sum of the $\alpha$th atomic orbital $\varphi_{\mathbf{R}\alpha}(\mathbf{r})$ at $\mathbf{R}$th cell:

$$\chi_\alpha^{\mathbf{k}}(\mathbf{r}) = \frac{1}{\sqrt{N}} \sum_{\mathbf{R}} e^{i\mathbf{k}\cdot\mathbf{R}} \varphi_{\mathbf{R}\alpha}(\mathbf{r}) \tag{5}$$

Here $N$ denotes the number of unit cells. The combination coefficients $C_\alpha^{n\mathbf{k}}$ are determined by solving the eigenvalue equation of the Hamiltonian

$$H |\psi_{n\mathbf{k}}\rangle = E_{n\mathbf{k}} |\psi_{n\mathbf{k}}\rangle \tag{6}$$

under the basis formed by $\chi_\alpha^{\mathbf{k}}$. This is equivalent to solve the eigenvalue problem of the Bloch Hamiltonian matrix $H(\mathbf{k})$ whose elements are given by

$$H_{\alpha\beta}(\mathbf{k}) = \langle \chi_\alpha^{\mathbf{k}} | H | \chi_\beta^{\mathbf{k}} \rangle \tag{7}$$

Next, we present the Hamiltonian matrices corresponding to the three types of lattices. We denote by $\epsilon$ the onsite potential. For the kagome lattice and breathing kagome lattice, choose lattice vectors as $\mathbf{a}_1 = (1, 0)$, $\mathbf{a}_2 = (\frac{1}{2}, \frac{\sqrt{3}}{2})$ and the reciprocal lattice vectors as $\mathbf{b}_1 = (2\pi, -\frac{2\pi}{\sqrt{3}})$, $\mathbf{b}_2 = (0, \frac{4\pi}{\sqrt{3}})$. The corresponding matrix representation of the Hamiltonian is:

$$H(\mathbf{k}) = \begin{bmatrix} \epsilon & -t_1 - t_2 e^{i\mathbf{k}\cdot(\mathbf{a}_2 - \mathbf{a}_1)} & -t_1 - t_2 e^{i\mathbf{k}\cdot\mathbf{a}_2} \\ -t_1 - t_2 e^{-i\mathbf{k}\cdot(\mathbf{a}_2 - \mathbf{a}_1)} & \epsilon & -t_1 - t_2 e^{i\mathbf{k}\cdot\mathbf{a}_1} \\ -t_1 - t_2 e^{-i\mathbf{k}\cdot\mathbf{a}_2} & -t_1 - t_2 e^{-i\mathbf{k}\cdot\mathbf{a}_1} & \epsilon \end{bmatrix} \tag{8}$$

In the case of the kagome lattice, $t_1 = t_2$.

For the chiral breathing diatomic-kagome lattice, choose lattice vectors as $\mathbf{a}_1 = (1, 0)$, $\mathbf{a}_2 = (-\frac{1}{2}, \frac{\sqrt{3}}{2})$ and the reciprocal lattice vectors as $\mathbf{b}_1 = (2\pi, \frac{2\pi}{\sqrt{3}})$, $\mathbf{b}_2 = (0, \frac{4\pi}{\sqrt{3}})$. The corresponding matrix representation of the Hamiltonian is:

$$H(\mathbf{k}) = \begin{bmatrix} \epsilon & -t_1 & -t_1 & 0 & -t_2 e^{i\mathbf{k}\cdot\mathbf{a}_2} & 0 \\ -t_1 & \epsilon & -t_1 & 0 & 0 & -t_2 e^{-i\mathbf{k}\cdot\mathbf{a}_1} \\ -t_1 & -t_1 & \epsilon & -t_2 & 0 & 0 \\ 0 & 0 & -t_2 & \epsilon & -t_3 & -t_3 \\ -t_2 e^{-i\mathbf{k}\cdot\mathbf{a}_2} & 0 & 0 & -t_3 & \epsilon & -t_3 \\ 0 & -t_2 e^{i\mathbf{k}\cdot\mathbf{a}_1} & 0 & -t_3 & -t_3 & \epsilon \end{bmatrix} \tag{9}$$

For comparison with the STM results, we computed the local density of states (LDOS) for these TB models on clusters in Supplementary Fig. 13. Notably, the bright triangles with enhanced LDOSs in d$I$/d$V$ maps correspond to the small triangles with larger $t$ in the TB simulations. The LDOS is evaluated by the formula[37]

$$\text{LDOS}(E, i) = \sum_n |\psi_n(i)|^2 \delta(E - E_n) \tag{10}$$

where $i$ labels a lattice site of the cluster, and the variable $E$ represents the energy. $E_n$ and $\psi_n$ are the eigenenergies and eigenstates of the TB Hamiltonian on the cluster, respectively. The $\delta$-function is approximated by the Lorentzian with a broadening of 0.1 eV.

## Data availability

All data needed to support the conclusions in the study are available within the article and/or the supplementary files. Data underlying Figs. 2–6 and Supplementary Figs. 1–13 are provided in the source data file with this paper. Source data are provided in this paper. Source data are provided with this paper.

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

## Acknowledgements

This work was supported by the Innovation Program for Quantum Science and Technology (No. 2021ZD0303302), the National Natural Science Foundation of China (22002149, 12074359), the Chinese Academy of Sciences (XDB36020200), the CAS Project for Young Scientists in Basic Research (YSBR-054), the Anhui Initiative in Quantum Information Technologies (AHY090000), the New Cornerstone Science Foundation, and the Fundamental Research Funds for the Central Universities. R.Y. acknowledges the support of the National Natural Science Foundation of China (12304235).

## Author contributions

C.M. and B.W. conceived the project. The on-surface synthesis and the STM/nc-AFM measurements were performed by R.Y., X.Z., Y.W., Y.L. and Z.W. under the supervision of C.M., S.T. and B.W. Z.-L.Q. synthesized and characterized the precursor molecule under the supervision of Y.-Z.T. The DFT calculations were carried out by Q.F. and B.L., and the TB simulations by T.H. and L.W. under the supervision of Z.F.W., W.H. and J.Y. R.Y., C.M. and B.W. wrote the manuscript with input from all authors.

## Competing interests

The authors declare no competing interests.
