## [Peer Review File · Nature Communications]

REVIEWER COMMENTS

Reviewer #1 (Remarks to the Author):

Ruoting Yin et al. reported on a systematic study and a generalized strategy to achieve self-assembled kagome lattices with variable kagome lattice parameters using on-surface chemistry. The STM, nc-AFM experiments and DFT calculations were used as methods and corroborative evidence to further support the conclusions. Three exemplar XHOF lattices were showcased as demonstrations, i.e., regular, breathing and chiral breathing diatomic kagome lattices, and their electronic properties were investigated and compared. Overall, this manuscript is well-written, logically structured and scientifically sound. The methodology of such generalized networking strategy proposed by the authors is quite noteworthy. However, I would like to raise the following comments/questions concerning this manuscript:

1. Typical Br or other halogen atoms appear as bright dots in nc-AFM due to their relatively large electronegativity. Such feature is not present in Fig. 2 nc-AFM images. Is this due to the charge transfer with substrates/surrounding molecules? What about the nc-AFM image of bare Ag/Au substrate with preexisting Br atoms? Does it show any bright dots at sites of Br atoms?
2. Another question following the previous one: elevated annealing temperature of 473 / 573 K for ~ 30 min should give the top layer Ag/Au atom enough kinetic energy to overcome the potential barrier popping out onto surface and become Ag/Au adatoms, which will supply an abundant number of free-moving Ag/Au adatoms on surface during annealing. Could the coordination atom be Ag/Au adatoms? Did the authors observe any different coordinator/linkage other than Br atoms?
3. I suppose the substrate is included in the electrostatic potential simulations in Fig. 3? If so, charge transfer is also taken into account. What is the reason for the discrepancy between calculated FB1/FB2 energies (Fig. 3f) and experimentally observed ones (Fig. 5d-e)? Also, the Au(111) dI/dV spectrum in Fig. 5b looks a bit weird, the characteristic -470 mV peak cannot be recognized at all. This could be a sign that the data set presented in Fig. 5 has been compromised due to an incorrect tip state.
4. In line 145, the “setpoint condition” should be explicitly clarified with a real-space spot, i.e., on bare Ag/Au, or on molecules.

In summary, I found the results in this manuscript solid and well-structured with some minor unsettled issues. The highlight of this manuscript is the methodology of such generalized networking strategy proposed by the authors and the limitless transferability thereupon. Therefore, I believe this manuscript

deserves to be published in Nature Communications, provided that the above-mentioned comments/questions are well addressed/resolved in a revised version manuscript.

Reviewer #2 (Remarks to the Author):

The paper describes the growth of Kagome lattices on Ag(111) by the deposition of organic precursors and subsequent annealing. The characterization by imaging with scanning tunneling microscopy (STM) and non-contact atomic force microscopy (nc-AFM) is impressive and of high quality. The authors also perform dI/dV -spectroscopy, where several peaks in range of $\pm 3V$ are observed. In comparison with tight-binding and DFT calculations, they relate some of the peaks to LUMO and HOMO levels and some to flatbands. The interpretation of these dI/dV spectra is not completely convincing. Therefore, I recommend major revision before publication.

The following points should be clarified:

-The authors describe that they observe the surface state of Ag(111) at $-63mV$. What is rather confusing is that the surface state seems still be present on the molecule covered areas. (see Fig. 4d) If the molecular lattice confines the surface state, the energy should be shifted upwards depending on the size of the pores. The original peak at $-63mV$ should disappear on the Kagome lattice and be observed at higher energies. This should be clearly pointed out.

-The identification of peaks to be related to flat bands, pore states or LUMO+2 seems a bit arbitrary. The width of peaks related to flat bands seems rather large. This is also a major point to be clearly addressed in a revised version.

Reviewer #3 (Remarks to the Author):

In this manuscript, Yin et al report flat bands (FB) formed in three artificial Kagome lattices using STM and STS. The artificial lattices are formed through self-assembly of three molecular precursors with Br atoms. Different structures provide regular, breathing, and chiral breathing diatomic-Kagome lattices that feature unique FBs.

The strategy of using molecules to form artificial lattices is not new and has been widely reported, including those featuring FB such as Lieb lattice and Kagome lattice. In this sense, this work lacks novelty. The major claim of this work is to use three molecules to build different Kagome lattices. However, the data interpretations are not well established, see detail comments below. Overall, this manuscript does not meet the standard of NC.

Comments:

1. Besides the regular Kagome structure, in the breathing, and chiral breathing diatomic-Kagome structures, the lattice sites are occupied by Br atoms. This contradicts the concept of using empty sites as artificial “atoms” to form artificial lattices. Thus, conceptually, to define these two structures as the proposed artificial Kagome lattices is wrong.
2. Even in the case of the regular Kagome lattice, the assignment of the “shoulder” in STS to a FB is not convincing. In addition, the molecule and Br sites show a clear peak at the same energy, which strongly indicates the “shoulder” feature has other origin instead of a FB.
3. To assign the “humps” in STS shown in Figures 4d and 5b is ambiguous. Such features can be attributed to other mechanisms, such as confinement of the surface states.
4. The enhanced STS signal at the edges shown in Figure 6 is an often observed phenomena and can be caused by other processes, not necessary indicating “topological edge states”.

Point-to-point reply to comments from Reviewer 1:

General Comments: Ruoting Yin *et al.* reported on a systematic study and a generalized strategy to achieve self-assembled kagome lattices with variable kagome lattice parameters using on-surface chemistry. The STM, nc-AFM experiments and DFT calculations were used as methods and corroborative evidence to further support the conclusions. Three exemplar XHOF lattices were showcased as demonstrations, *i.e.*, regular, breathing and chiral breathing diatomic kagome lattices, and their electronic properties were investigated and compared. Overall, this manuscript is well-written, logically structured and scientifically sound. The methodology of such generalized networking strategy proposed by the authors is quite noteworthy. However, I would like to raise the following comments/questions concerning this manuscript:

Reply to General Comments 1: We thank the Reviewer for the positive and insightful comments of our manuscript.

Comment C1: Typical Br or other halogen atoms appear as bright dots in nc-AFM due to their relatively large electronegativity. Such feature is not present in Fig. 2 nc-AFM images. Is this due to the charge transfer with substrates/surrounding molecules? What about the nc-AFM image of bare Ag/Au substrate with preexisting Br atoms? Does it show any bright dots at sites of Br atoms?

Reply to C1: We thank the Reviewer for pointing this out. Indeed, the Br atoms are almost invisible in the nc-AFM images with a far distance (**Fig. 2c,h,m**), and only appear as slightly bright dots in those zoom-in and closer-tip nc-AFM images (**Fig. 2d,i,n**), as marked by red circles and red arrows (see **Revised Fig. 2**). To better present these features of the Br atoms, we show the raw nc-AFM images of **Fig. 2d,i,n**, without superimposed structural models and markers, as a **new Supplementary Fig. 2 (also attached below)**. Features related to Br atoms and the Br \cdots H bonds between molecules can be more clearly seen.

In the revision, we have clarified this point in **the caption of Fig. 2**. It reads

“... In **d**, **i**, and **n**, the Br \cdots H bonds and the positions of Br atoms highlighted by the red dashed lines and the red circles/arrows, with the raw images shown in Supplementary Fig. 2.”

New Supplementary Fig. 2

Following the Reviewer’s comments, we also have looked at the nc-AFM images of the preexisting Br atoms. In our experiment, the Br atoms were achieved by breaking the C-Br bonds during on-surface polymerization of 4BrPn (6,13-bis(dibromomethylene)-6,13-dihydropentacene), following a previous report (Cirera *et al.*, Nat. Nanotechnol. 15, 437 (2020)). As shown in **Reply Fig. 1a**, the STM image acquired at 5 K clearly presents the Br atoms stitching to the polymer edges (white dashed circles), slightly away from the polymer edges (white and red arrows), and between polymer chains (white dashed rectangles). The nc-AFM image acquired at the same area is shown in **Reply Fig. 1b**, where only Br atoms away from the polymer edges are clearly seen as bright dots (white

and red arrows), while those adsorbed at the individual or closely-packed polymer edges just display the shallow contrast (white dashed circles and rectangles). These results can suggest that the charge transfer between the Br atoms and the surrounding molecules may affect the contrast of the adsorbed Br atoms, in line with the Reviewer's expectation.

In the revision, we have added a sentence to discuss this point on **Page 7**. It reads

“Considering the relatively large electronegativity of the Br atom, its brightness in the nc-AFM images is quite low, suggesting charge transfer with the surrounding molecules.”

Reply Fig. 1 | Contrasts of Br atoms at different environments in nc-AFM images. **a**, STM image, and **b**, the corresponding nc-AFM image of polymer chains that were grown by depositing the precursor 4BrPn (6,13-bis(dibromomethylene)-6,13-dihydropentacene) on an Au(111) surface and subsequent annealing of the surface at 390 K.

Comment C2: Another question following the previous one: elevated annealing temperature of 473 / 573 K for ~ 30 min should give the top layer Ag/Au atom enough kinetic energy to overcome the potential barrier popping out onto surface and become Ag/Au adatoms, which will supply an abundant number of free-moving Ag/Au adatoms on surface during annealing. Could the coordination atom be Ag/Au adatoms? Did the authors observe any different coordinator/linkage other than Br atoms?

Reply to C2: We thank for the Reviewer's comment. Indeed, at the elevated annealing temperature of 473/573 K, there will be an abundant number of free-moving Ag/Au adatoms on surface. It can be evidenced by the formation of organometallic intermediates, such as the organometallic trimers **M-C66** on Ag(111) (Fig. 2f). During this process, the metal adatoms can form C-metal bonds after the breaking of C-halogen bonds in an Ullmann-like reaction.

However, in our experiment, we found that without the preexisting Br atoms, benzene molecules did not self-assemble into islands. Besides, our previous work demonstrated that the **C66/Br/Au(111)** superlattice will disassemble after removal of Br atoms by annealing at 670 K (ref. 52). These results can confirm the coordination atoms are Br in the XHOFs.

Comment C3: I suppose the substrate is included in the electrostatic potential simulations in Fig. 3? If so, charge transfer is also taken into account. What is the reason for the discrepancy between calculated FB1/FB2 energies (Fig. 3f) and experimentally observed ones (Fig. 5d-e)? Also, the Au(111) dI/dV spectrum in Fig. 5b looks a bit weird, the characteristic -470 mV peak cannot be recognized at all. This could be a sign that the data set presented in Fig. 5 has been compromised due to an incorrect tip state.

Reply to C3: We thank the Reviewer for pointing these out. We are sorry for the misleading about the electrostatic

potential simulations and the band structure calculations. The electrostatic potential simulated in Fig. 3a,c,e were done by density functional theory (DFT) calculations, where the substrates that contain four metal atomic layers were included and thus the charge transfer was taken into account. Differently, the electronic bands in Fig. 3b,d,f were calculated by the tight-binding (TB) models by considering different electron hopping strengths between atomic orbitals associated to lattice sites, where energies of the bottom bands were manually aligned to zero (not Fermi level). So, the flat band energies can only be qualitatively compared with the experimental ones.

While the DFT-simulated electrostatic potential landscapes display the characteristic patterns of the kagome lattice (Fig. 3a), the breathing kagome lattice (Fig. 3c), and the chiral breathing diatomic-kagome lattice (Fig. 3e), the electronic band structures based on the same DFT method and structural models are dominated by bands from the substrate atoms, as shown in the **Reply Fig. 2 (also attached below)**. It is impossible to extract the band information related to the artificial electronic kagome lattices. What's more, the semi-finite substrates including only four atomic layers, due to the computational feasibility, could not effectively reproduce the accurate surface states of the bulk crystal (I. N. Yakovkin, *Comput. Mater. Sci.* 156, 84 (2019)). This is the reason why we adopted the TB band structures of the artificial electronic kagome lattices in Fig. 3. Similar treatment was also adopted by ref. 47, due to the same reason. Well in line with our TB calculations, our experimental results present the main features of the corresponding kagome lattices.

To clarify this point, in the revision, we have also revised following sentences. They read

On **Page 7**, “To further shed light on our strategy for the electronic kagome lattices, we simulate the electrostatic potentials of the three superlattices (Fig. 3a,c,e) using the DFT method based on the structural models obtained in Fig. 2.”

On **Page 8**, “Considering our DFT calculations that include semi-infinite metal surfaces with only four metal-atom layers, due to a computational feasibility, cannot accurately describe the electronic structures of the Shockley SSs⁵⁰ and the confined states⁴⁷ in the XHOF superlattices, we calculate the band structures of the three electronic kagome lattices by adopting the TB theory that involves the electron hopping strength between atomic orbitals associated to lattice sites.”

Reply Fig. 2

About the comment on the spectra in **Fig. 5b**, we want to clarify that, different from other dI/dV curves, they were normalized by $dI/dV/(I/V)$ to suppress the strong intensity of the LUMO+2 orbital. This may be reason why the Au(111) spectrum looks a bit weird. As show in the **revised Supplementary Fig. 9 (also attached below)**, we put the pristine dI/dV spectra (**Fig. 9d**) and the normalized $dI/dV/(I/V)$ spectra (**Fig. 9e**) together. The dI/dV spectrum of the Au(111) substrate shows a clear bump at around -450 mV, correspondingly to the characteristic Au(111) surface state, which should suggest the correct tip status.

In the revision, we have rearranged **Fig. 5 (attached below)** to enhance the electronic features, and revised following sentences to address this point **on Page 11**. It reads

“...Here, the dI/dV spectra are normalized by $(dI/dV)/(I/V)$ to suppress the strong intensity of the LUMO+2 orbital (Supplementary Fig. 9). In addition, the energy upward shift of Au(111) SSM from -0.45 eV to -0.27 eV in the C66/Br/Au(111) superlattice is also more clearly displayed (Fig. 5b and Supplementary Fig. 9), which is in line with the modified boundary conditions⁵³⁻⁵⁵.”

Revised Supplementary Fig. 9

Revised Fig. 5

Comment C4: In line 145, the “setpoint condition” should be explicitly clarified with a real-space spot, i.e., on bare Ag/Au, or on molecules.

Reply to C4: In the revision, we have clarified this point in the caption of Fig. 2. It reads “..., with respect to the setpoint condition $V_s = -0.8$ V, $I_t = 10$ pA on molecules.”

Summarized Comments: In summary, I found the results in this manuscript solid and well-structured with some minor unsettled issues. The highlight of this manuscript is the methodology of such generalized networking strategy proposed by the authors and the limitless transferability thereupon. Therefore, I believe this manuscript deserves to be published in Nature Communications, provided that the above-mentioned comments/questions are well addressed/resolved in a revised version manuscript.

Reply to Summarized Comment: We thank the reviewer for highlighting the importance of our manuscript, for the insightful comments to improve our manuscript, and for the recommendation of publication of our manuscript in Nature Communications.

Point-to-point reply to comments from Reviewer 2:

General Comments: *The paper describes the growth of Kagome lattices on Ag(111) by the deposition of organic precursors and subsequent annealing. The characterization by imaging with scanning tunneling microscopy (STM) and non-contact atomic force microscopy (nc-AFM) is impressive and of high quality. The authors also perform dI/dV-spectroscopy, where several peaks in range of $\pm 3V$ are observed. In comparison with tight-binding and DFT calculations, they relate some of the peaks to LUMO and HOMO levels and some to flatbands. The interpretation of these dI/dV spectra is not completely convincing. Therefore, I recommend major revision before publication.*

Reply to General Comments: We thank the Reviewer for appreciation of the high quality of our data. As for the interpretation of the dI/dV spectra is not completely convincing, we have revised our manuscript to more clearly interpret these spectra.

The following points should be clarified:

Comment C1: *The authors describe that they observe the surface state of Ag(111) at -63mV. What is rather confusing is that the surface state seems still be present on the molecule covered areas. (see Fig. 4d) If the molecular lattice confines the surface state, the energy should be shifted upwards depending on the size of the pores. The original peak at -63mV should disappear on the Kagome lattice and be observed at higher energies. This should be clearly pointed out.*

Reply to C1: We thank the Reviewer for pointing this out and we are sorry for the confusing description in the previous version. In order to make this point clear, we have **revised Fig. 1 (also attached below)** by adding a schematic drawing of the mechanism of the formation of artificial electronic kagome bands. It can be seen now that the kagome bands are formed by reshaping *the dispersive band of the Shockley surface state (SS)* from the patterning effect of the XHOFs. The dispersive Shockley SS exhibits its minimum (SSM) below E_F , which is at around -100 mV for bare Ag(111) surface and -450 mV for the Au(111) surface, respectively.

In the previous version of our manuscript, we generally called *the dispersive Shockley surface state (SS)* and *the Shockley SS minimum (SSM)* as the surface state, which caused the confusing. In the revision, we have distinguished these two terms all through the manuscript, and have further described the strategy of the formation of artificial electronic kagome lattices. They read

On **Page 9**, “As a reference, the Shockley SS on the Ag(111) surface displays a characteristic step-like feature in the dI/dV spectrum with the minimum (SSM) at around -80 mV.”

On **Page 5**, “Our strategy for designing artificial electronic kagome lattices uses (quasi)hexagonal organic building blocks that self-assemble into XHOFs through halogen-based hydrogen bonds (Fig. 1a). This is accomplished by the templating effect of the XHOFs on coinage metals¹⁻³, which can regulate the movement of the confined Shockley surface-state electrons within the triangular empty regions surrounded by halogen atoms and organic building blocks, as marked by blue shadowing in Fig. 1a, to satisfy the conditions of emerging artificial kagome lattices. The phase cancellation in the kagome lattices renders the kagome FBs and real-space localization, which reshapes the dispersive band of the Shockley surface state (SS) in the coinage metal surface to diverse kagome bands (Fig. 1b).”

In our experiment, the Shockley SS minima can be present on the molecule covered areas with slightly modified energies and the kagome flat bands are observed above E_F . These results suggest that the XHOF superlattices just slightly modified the band bottom of the dispersive Shockley surface state. This is consistent with a lot of previous reports of adsorbed molecular systems on coinage metals, where energies of the SSMs are just slightly changed due

to the modified boundary conditions in the presence of adsorbates (cf. Nicoara, N. *et al.*, *Org. Electron.* 2006, 7, 287–294; Malterre, D. *et al.*, *New J. Phys.* 2007, 9, 391; Park, J.-Y. *et al.*, *Phys. Rev. B* 2000, 62, R16341–R16344; Ruffieux, P. *et al.*, *ACS Nano* 2012, 6, 6930–6935; Ma, C. *et al.*, *Nano Lett.* 2017, 17, 6241; Yin R. *et al.*, *J. Am. Chem. Soc.* 2022, 144, 14798–14808).

In the revision, we have clarified this point in the paragraphs related to **Figs. 4 and 5**, respectively.

On **Page 9–10**, “The **M-C66**/Br/Ag(111) superlattice presents richer electronic states in the dI/dV spectra (Fig. 4d and Supplementary Fig. 6). The Ag(111) SSM is consistently observed over the superlattice, with slightly different energies and shapes compared to those of the SSM on the pristine surface, suggesting that the modification of the band bottom of the dispersive Shockley SS is small in the presence of the XHOF.”

On **Page 11**, “In addition, the energy upward shift of Au(111) SSM from -0.45 eV to -0.27 eV in the **C66**/Br/Au(111) superlattice is also more clearly displayed (Fig. 5b and Supplementary Fig. 9), which is in line with the modified boundary conditions⁵³⁻⁵⁵.”

Revised Fig. 1

Comment C2: The identification of peaks to be related to flat bands, pore states or LUMO+2 seems a bit arbitrary. The width of peaks related to flat bands seems rather large. This is also a major point to be clearly addressed in a revised version.

Reply to C2: We thank the Reviewer for pointing these out.

In the revision, we have added the detailed analyses of the experimental data related to the flat bands, pore states and the LUMO+2 orbital, by providing new data of the DFT-simulated LDOS maps corresponding to the molecular orbitals in all three superlattices in the **revised Supplementary Figs. 4, 7, and 8 (also attached below)**. For the case of the **C66**/Br/Au(111) superlattice (**Supplementary Fig. 8**), compared to the LUMO+2 peak, the LUMO+1 peak is much weaker and close to the LUMO peak, showing as a shoulder-like feature of the LUMO peak in the PDOS curve (**Supplementary Fig. 8a**). After directly comparing the simulated PDOS curve and LDOS maps with the experimental ones, we can reasonably assign the experimentally observed strong peak at 2.57 eV to the LUMO+2 orbital of **C66**, which is also in line with our previous observations of the isolated **C66** nanographenes (Yin *et al.*, *J. Am. Chem. Soc.* 144, 14798 (2022)). For the case of the **M-C66**/Br/Ag(111) superlattice (**Supplementary Fig. 7**),

after comparing the experimentally observed peak intensity and spatial distribution at 2.50 V with the theoretically simulated unoccupied orbitals, we can find the resemblance with the LUMO+3 orbital of the **M-C66** molecule. Thus, we have changed the “pore state” in the previous version to the “LUMO+3” in the revision, which gives the same nomenclature for both structures. The LUMO+3 orbital has also been superimposed on the experimental map in the **revised Fig. 4i (attached below)**.

After assigning the molecular orbitals, the other emerging peaks within the molecular HOMO–LUMO gaps show clear localization at the intervals between the molecular structures. The corresponding dI/dV maps well resemble the patterns of the electronic kagome lattices with different varieties. On the other hand, the emerging peaks are also in line with the TB predicted number of flat bands in different kagome lattices. Therefore, we can safely assign the flat bands.

In the revision, we have clarified the above points by making following changes.

On **Page 10**, “In addition, one can see that the FBs are significantly suppressed at the central hole of the **M-C66** structure, where a strong LDOS enhancement is observed at approximately 2.50 eV (Fig. 4d,i). It can be safely assigned to the forth lowest unoccupied molecular orbital (LUMO+3) of the **M-C66**, after compared with the simulated spatial distributions of various molecular orbitals (Supplementary Fig. 7).”

On **Page 11**, “By combining dI/dV measurements (Fig. 5a,b) and DFT calculations (Supplementary Fig. 8) of the **C66/Br/Au(111)** superlattice, we can assign the HOMO at -0.87 eV (Fig. 5c), LUMO at 1.85 eV (Fig. 5f), and LUMO+2 at 2.57 eV (Fig. 5g), which are also consistent with our previous observations of the isolated **C66** nanographenes⁵².”

On **Page 12**, “The emergence of electronic kagome lattices relies on two recipes. One is the potential barriers induced by the molecular nanostructures, which means that the emerging kagome bands should reside in the HOMO–LUMO gaps, in line with the experimental observations (Figs. 3, 4, and 5). The other is the charge transfer from metal substrates to molecular structures in the XHOFs, which consistently induces charge ordering, as exhibited by the calculated electrostatic potentials (Fig. 3a,c,e).”

About peak widths of the flat bands, we agree with the Reviewer that they seem rather large, ranging from about 0.2 to 0.4 eV in our study, considering the full width at half peak width. However, we find these values are comparable with the peak widths of the molecular orbital energy levels observed simultaneously, which are supposed to be extremely narrow. These features are similar to a previous report on the formation of an electronic chiral kagome-honeycomb lattice on Cu(111) (ref. 47). The broadening effect can be understood as the limited coherent lifetime of the surface states resulting from the surface-bulk scattering of the electrons on the metallic surface (A. Eiguren. *et al.*, Phys. Rev. Lett. 2002, 88, 066805; Saoirse E. Freeney *et al.*, ACS Nanosci. Au 2022, 2, 198–224).

In the revision, we have addressed this point on **Page 15**. It reads

“We also need to admit that both our strategy and the tip-assisted manipulation approaches suffer from the broadening effect of the kagome flat bands due to the surface-bulk scattering of electrons that results in the limited coherent lifetime on the metallic surface^{3,61}, as observed in our experiment with finite FB peak width ranging from about 0.2 to 0.4 eV and in previous reports with similar values^{41,47}.”

Revised Supplementary Fig. 4

Revised Supplementary Fig. 7

Revised Supplementary Fig. 8

Revised Fig. 4

Point-to-point reply to comments from Reviewer 3:

General Comments: *In this manuscript, Yin et al report flat bands (FB) formed in three artificial Kagome lattices using STM and STS. The artificial lattices are formed through self-assembly of three molecular precursors with Br atoms. Different structures provide regular, breathing, and chiral breathing diatomic-Kagome lattices that feature unique FBs. The strategy of using molecules to form artificial lattices is not new and has been widely reported, including those featuring FB such as Lieb lattice and Kagome lattice. In this sense, this work lacks novelty. The major claim of this work is to use three molecules to build different Kagome lattices. However, the data interpretations are not well established, see detail comments below. Overall, this manuscript does not meet the standard of NC.*

Reply to General Comments: The Reviewer summarized our work, however, with full respect, we disagree with the comment that *this work lacks novelty*. The Reviewer mainly concerned the similarity of current work with some others. We would like to highlight the novelty of the current study, and have revised our manuscript to clarify the importance of our findings.

Firstly, despite the strategy of using molecules to form artificial lattices including FB feature has been reported before, however, they were generally obtained by atom-by-atom manipulation with an STM tip. Indeed, while STM manipulation provides an ultimate control over the lattice topology, its drawbacks, including being time-consuming, laborious, inefficient, and lacking scalability, make it unsuitable for practical applications. Therefore, a simple and effective methodology of generalized networking strategy is highly desired. In this work, we devise a general strategy to reliably and effectively construct varieties of electronic kagome lattices by utilizing the on-surface synthesis of halogen hydrogen-bonded organic frameworks (XHOFs). Our approach could surpass those drawbacks of the STM tip-assisted manipulation to generate variety of electronic kagome lattices with huge complexity, such as the chiral breathing diatomic-Kagome lattice that was not realized before, and to achieve large domain sizes of the kagome lattices for up to 100 nm or more (Supplementary Fig. 1).

Secondly, our strategy can be seamlessly combined with the broadly interested on-surface chemistry to controllably generate a new type of two-dimensional frameworks, i.e., the XHOFs, through a novel self-assembly mechanism by halogen hydrogen-bonding. This offers the almost limitless tunabilities for the symmetries and peripheral shapes of the organic building blocks in the XHOFs, which enables tremendous varieties of the artificial kagome lattices. The XHOFs represent a new type of frameworks, which also open a door to assemble the atomically precise on-surface synthesized nanostructures for realizing not only collective properties but also practical multifunctionalities, considering the porous 2D structures that can be periodically embedded with individual or trimmerized metal atoms. Following this strategy, one can also achieve various artificial electronic lattices, such as the Lieb lattice and fractal lattices, by varying the symmetries of the building blocks and metal substrates. As pointed out by **Reviewer 1**, *“The highlight of this manuscript is the methodology of such generalized networking strategy proposed by the authors and the limitless transferability thereupon.”*

In the revision, we have summarized above information as a new paragraph in the section of **Discussion**. It reads

“The physical mechanism underlying our strategy for creating artificial electronic kagome lattices is similar to the artificial electronic lattices patterned by CO molecules on metal surface via STM tip-assisted manipulation³⁶⁻⁴¹. Notably, while the STM manipulation provides an ultimate control over the lattice topology, it has obvious drawbacks, including being time-consuming, laborious, inefficient, and lacking scalability. However, our approach could surpass those drawbacks to generate variety of electronic kagome lattices with huge complexity, such as the chiral breathing diatomic-kagome lattice that was not realized before, and to achieve large domain sizes of the kagome lattices for up to 100 nm or more (Supplementary Fig. 1). Another advantage of our strategy lies in the fact that our strategy can be

further seamlessly combined with the well-developed on-surface chemistry. This offers the almost limitless tunability for the symmetries and peripheral shapes of the organic building blocks in the XHOFs, which enables tremendous varieties of the artificial kagome lattices. These exciting aspects offer great opportunities for simulating and exploring the electronic and topological properties of complex quantum materials. The 2D XHOFs also open a door to assemble the atomically precise on-surface synthesized nanostructures for realizing not only collective properties but also practical multifunctionalities⁵⁷⁻⁵⁹, considering the porous structures that can be periodically embedded with individual or trimerized metal atoms and other functional groups⁶⁰.”

We believe that the present strategy that combines the new-type frameworks of XHOFs for producing electronic kagome lattices is novel and that the on-surface chemistry for the formation of XHOFs is essentially important, which should meet the high standard of the Nature Communications. We sincerely hope that the Reviewer may find our revised manuscript has been substantially improved in better presentation of the novelty and our data interpretations.

***Comment C1:** Besides the regular Kagome structure, in the breathing, and chiral breathing diatomic-Kagome structures, the lattice sites are occupied by Br atoms. This contradicts the concept of using empty sites as artificial “atoms” to form artificial lattices. Thus, conceptually, to define these two structures as the proposed artificial Kagome lattices is wrong.*

Reply to C1: We thank the Reviewer for the question and we are sorry that we did not present the concept of our strategy clearly in the previous version. In the revision, we have **revised Figs. 1 and 2 (also attached below)** to better illustrate our design concept.

As illustrated in **Fig. 1a**, our strategy is accomplished by the templating effect of the XHOFs on coinage metals. The potential barriers formed by halogen atoms and organic building blocks generate triangular regions of the empty metal surfaces, as marked by blue shadowing, which can regulate the movement of the confined Shockley surface-state electrons within the triangles and emerge artificial kagome lattices. The quantum destructive interference of electrons within the triangles can effectively convert the dispersive band of the Shockley surface state (SS) into kagome bands, as schematically illustrated in **Fig. 1b**.

Following this design concept, in the benzene/Br/Ag(111) superlattice of **Fig. 2e**, the empty regions formed by Br atoms and benzene molecules, as marked by yellow shadowing, will form the regular kagome lattice. Due to the small size of the Br atom, the superposition of the confined states at the three nearby sites will create significantly enhanced charge density distributions within the triangles of the kagome lattice, as marked in **Fig. 2e**. This expectation is directly confirmed by the dI/dV maps acquired at the flat band energy range in **Fig. 3c** and **Supplementary Fig. 4**, which show triangular patterns, substantiating the correctness of our concept. Then, similarly, we can draw the triangles formed by Br atoms and molecular carbon nanostructures in the **M-C66/Br/Ag(111)** superlattice of **Fig. 2j** and in the **C66/Br/Au(111)** superlattice of **Fig. 2o**. One can easily see that these triangles correspondingly form the breathing kagome lattice and the chiral breathing diatomic-kagome lattice. Therefore, the definition of all three artificial kagome lattice is following the same concept and it should be correct.

Besides, our calculated electrostatic potential maps, TB band structures, and experimental electronic measurements for all three artificial kagome lattices are consistent. All information together can affirm that the definition of the three artificial kagome lattices is correct.

In the revision, we have included the clarification of our concept as an individual paragraph on **Page 5**, and revised the description of the formation of different kagome lattices in **revised Fig. 2 on Page7**. They read

On **Page 5**, “Our strategy for designing artificial electronic kagome lattices uses (quasi)hexagonal organic building blocks that self-assemble into XHOFs through halogen-based hydrogen bonds (Fig. 1a). This is

accomplished by the templating effect of the XHOFs on coinage metals¹⁻³, which can regulate the movement of the confined Shockley surface-state electrons within the triangular empty regions surrounded by halogen atoms and organic building blocks, as marked by blue shadowing in Fig. 1a, to satisfy the conditions of emerging artificial kagome lattices. The phase cancellation in the kagome lattices renders the kagome FBs and real-space localization, which reshapes the dispersive band of the Shockley surface state (SS) in the coinage metal surface to diverse kagome bands (Fig. 1b).”

On **Page 7**, “The hexagonal XHOF superlattices (Fig. 2b,g,l) can effectively pattern the surface potential landscapes with the formation of triangular regions of the exposed metal surfaces, as depicted in Fig. 2e,j,o by yellow/blue shadowing, which are defined by the surrounding Br atoms and the molecular structures in the presence of Br⋯H bond networks. Following our design strategy, these XHOF superlattices should give rise to different varieties of electronic kagome lattices. ...”

Revised Fig. 1

Revised Fig. 2

Comment C2: Even in the case of the regular Kagome lattice, the assignment of the “shoulder” in STS to a FB is not convincing. In addition, the molecule and Br sites show a clear peak at the same energy, which strongly indicates the “shoulder” feature has other origin instead of a FB.

Reply to C2: In the previous version, the results of the regular kagome lattice were shown as Fig. 4a–c, which are presented as a new Supplementary Fig. 5 (also attached below) in the revision. The reference dI/dV spectrum from the bare Ag(111) surface does not clearly display the characteristic step-like feature of the Shockley surface state at around -100 mV, suggesting the tip was not clean. To exclude the possibility that the peak observed all over the superlattice was due to the special tip, we have carefully re-performed the experiment. The new data are shown in the Revised Fig. 4a–c, with more details in a new Supplementary Fig. 3 (both attached below). The new reference Ag(111) dI/dV spectrum shows the standard step-like Shockley surface state at around -80 mV, confirming a clean tip. At the meantime, the dI/dV spectra acquired in the benzene/Br/Ag(111) superlattice suggest a significant enhancement of the local density of states (LDOS) at an energy of 1.46 eV, with comparable intensities at the intervals between Br atoms or benzene molecules (magenta) and on the Br atom (green), but slightly weaker on the benzene molecule (blue) (Supplementary Fig. 3). The dI/dV map acquired at 1.35 V, corresponding to the rising edge of the FB peak to increase the contrast between different sites, displays the kagome lattice with enhanced LDOSs at the nonoccupied intervals (Fig. 4c and Supplementary Fig. 3), while the lattice disappears at a slightly lower energy of 1.00 eV (Fig. 4b). These results are consistent with previous data (Supplementary Fig. 5), suggesting that both results are correct and have the same origin of a kagome FB.

The observable FB intensity on the benzene molecule (Br atom) can be attributed to the penetration of confined electrons due to the small size of the benzene molecule (Br atom). This may reduce the confinement and broaden the FB bandwidth, which can be improved by adopting larger building blocks. As shown in the M-C66/Br/Ag(111) superlattice (Fig. 4d), the FB peaks are significantly suppressed at the center of the M-C66 structure. Nevertheless,

this proof-of-concept result confirms the feasibility of our design strategy.

In the revision, we have made the corresponding changes to clarify this point. They read

On **Page 9**, “The dI/dV spectra in Fig. 4a acquired in the benzene/Br/Ag(111) superlattice suggest a significant enhancement of the local density of states (LDOS) at an energy of 1.46 eV, which show comparable intensities at the intervals between Br atoms or benzene molecules (magenta) and on the Br atom (green), but slightly weaker on the benzene molecule (blue) (Supplementary Fig. 3). As a reference, the Shockley SS on the Ag(111) surface displays a characteristic step-like feature in the dI/dV spectrum with the minimum (SSM) at around -80 mV. According to the DFT simulations (Supplementary Fig. 4), the bandgap of the benzene molecule should be around 5 eV, with the highest occupied molecular orbital (HOMO) and the lowest unoccupied molecular orbital (LUMO) at -2.29 and 2.67 eV, respectively, which suggest that the observed intense peak at 1.46 eV should not originate from the orbitals of the benzene molecule, but from the significantly enhanced effective mass of the confined Shockley SS, such as the emerging kagome flat band. In line with this assignment, the dI/dV map acquired at 1.35 V, corresponding to the rising edge of the FB peak to increase the contrast between different sites⁵¹ (Supplementary Fig. 3), displays the kagome lattice with enhanced LDOSs at the nonoccupied intervals (Fig. 4c), while the lattice disappears at a slightly lower energy of 1.00 eV (Fig. 4b). Similar results are also obtained with a slightly different tip (Supplementary Fig. 5). The observable FB intensity on the benzene molecule results from the penetration of confined electrons due to the small size of the benzene molecule. This may reduce the confinement and broaden the FB bandwidth, which can be improved by adopting larger building blocks, as shown in the following. Nevertheless, this proof-of-concept result confirms the feasibility of our design strategy.”

On **Page 10**, “In addition, one can see that the FBs are significantly suppressed at the central hole of the **M-C66** structure, where a strong LDOS enhancement is observed at approximately 2.50 eV (Fig. 4d,i).”

Revised Fig. 4

New Supplementary Fig. 3

New Supplementary Fig. 5

Comment C3: To assign the “humps” in STS shown in Figures 4d and 5b is ambiguous. Such features can be attributed to other mechanisms, such as confinement of the surface states.

Reply to C3: Actually, our proposed mechanism for the formation of kagome lattices and the associated kagome bands is due to the confinement of the surface states due to the modified surface potential landscapes, as depicted in **Revised Fig. 1** (see **Reply to C1 of the same Reviewer**). However, we need to clarify that here the surface state should correspond to the dispersive band of the Shockley surface state (SS), which is different from the band minimum of the Shockley surface state (SSM), located at about -100 mV for the Ag(111) surface and -450 mV for the Au(111) surface, respectively (also refer to **Reply to C1 from Reviewer 2**). The pristine dispersive Shockley SS in the dI/dV spectrum shows a step-like enhancement of density of states at the SSM, together with a continuous and featureless background, as observed in the grey curve of **Revised Fig. 4a**. In our experiment, the Shockley SSM can be present on the molecule covered areas with slightly modified energies and shapes (**Fig. 4d**), as schematically illustrated in **Revised Fig. 1b**. This is consistent with a lot of previous reports of adsorbed molecular systems on coinage metals, where the energies of the SSM are just slightly changed due to the modified boundary conditions in the presence of adsorbates (cf. Nicoara, N. *et al.*, *Org. Electron.* 2006, 7, 287–294; Malterre, D. *et al.*, *New J. Phys.* 2007, 9, 391; Park, J.-Y. *et al.*, *Phys. Rev. B* 2000, 62, R16341–R16344; Ruffieux, P. *et al.*, *ACS Nano* 2012, 6, 6930–6935; Ma, C. *et al.*, *Nano Lett.* 2017, 17, 6241; Yin R. *et al.*, *J. Am. Chem. Soc.* 2022, 144, 14798–14808).

Then, the dI/dV peaks/pumps within the molecular HOMO–LUMO gaps suggest the significantly enhanced effective masses of the confined Shockley surface states at these energies. This feature is in line with the formation of kagome flat bands, which are further confirmed by the experimental dI/dV maps that show the breathing and chiral breathing diatomic-kagome lattices with obvious real-space localization of the densities of states.

In the revision, we have clarified this point by revising the sentences on **Page 9–10**. It reads

“The **M-C66/Br/Ag(111)** superlattice presents richer electronic states in the dI/dV spectra (Fig. 4d and Supplementary Fig. 6). The Ag(111) SSM is consistently observed over the superlattice, with slightly different energies and shapes compared to those of the SSM on the pristine surface, suggesting that the modification of the band bottom of the dispersive Shockley SS is small in the presence of the XHOF. On the **M-C66**, the HOMO and LUMO are found at -1.70 and 1.90 eV, respectively, which are confirmed by the corresponding dI/dV maps and simulated LDOS maps (Fig. 4e,f and Supplementary Fig. 7). At the inequivalent 1Br and 4Br sites, we observe two different peaks at 1.00 and 1.26 eV, respectively, within the molecular HOMO–LUMO gap, suggesting the greatly enhanced effective masses of the confined Shockley surface states at these energies due to the formation of kagome flat bands. Consistently, the dI/dV map at 1.00 eV (FB) reveals the breathing kagome lattice (Fig. 4g), as expected from our design model, where the LDOS intensities are inverse to the local Br concentrations.”

Comment C4: The enhanced STS signal at the edges shown in Figure 6 is an often observed phenomena and can be caused by other processes, not necessary indicating “topological edge states”.

Reply to C4: We want to clarify that there have two types of edges in our study. One is the edges of superlattice islands, as indicated by blue dotted lines in Fig. 6. The other is the edges of the artificial electronic kagome lattices, which consist of the outermost sites of the complete kagome lattices. It is true that for the first kind of edges, the physical boundary of a real material can host trivial boundary states, such as the dangling-bond states, which can contribute to the enhance STS signals. Our observations at peripheries of the two XHOF islands on Ag(111), i.e., the benzene/Br/Ag(111) superlattice (Fig. 6b) and **M-C66/Br/Ag(111)** superlattice (Fig. 6f), correspond to the first type. However, for the second type of edges, those artificial electronic kagome lattices with complete structures have no physically real chemical bonds to break. Therefore, they can only result from the emerging electronic states. This

is the case of the observed edge states for particular edge configurations in the experimentally determined breathing kagome lattice (Fig. 6f) and the chiral breathing diatomic-kagome lattice (Fig. 6j). Besides, our TB calculations of cluster models (Fig. 6g,k) confirm the presence of edge states with similar spatial distributions to the experiment.

In the revision, to further confirm the topological origin of the edge states in the breathing kagome lattice and the chiral breathing diatomic-kagome lattice, we have **revised Supplementary Fig. 12 (attached below)** to show that the observed edge states reside in the gap regions. The present experimental and theoretical results of the edge states in the gaps are consistent with the prediction of topological properties of the breathing kagome lattices.

Accordingly, on **Page 13–14**, one sentence has been added. It reads

“While the presence of hybridization states scattered around the peripheries of the superlattice islands (blue dotted lines in Fig. 6b,f) may disturb the above assignments, the absence of hybridization states at the **C66/Br/Au(111)** superlattice periphery provides undoubted evidence of the presence of topological edge states in the chiral breathing diatomic-kagome lattice (Fig. 6j–l). In line with these assignments and previous theoretical predictions^{15,18,41}, our TB calculations confirm the topological edge states emerging in the gap regions of the breathing kagome lattices (Supplementary Fig. 12).”

Revised Supplementary Fig. 12

REVIEWER COMMENTS

Reviewer #1 (Remarks to the Author):

The authors have satisfactorily addressed my critical issues/questions. The present version manuscript and SI have substantially improved in quality and clarity. However, I am not fully convinced by the authors' argument on the wired bare Au(111) spectrum in previous Fig. 5d (now Fig. 5a). Obviously one could see some disputable oscillations around Fermi. Comparing to the amplitude of the SS around -450 mV, the amplitude of these oscillations (two humps and two dips) are considerably large, which cannot be neglect at all. I assume the overall tip states in this data set is disputable/incorrect, which, as I commented last time, has inevitably compromised all the other dI/dV spectra (and mapping) data. It is suggested that the authors look for other data set with better tip states (if any), to eliminate possible controversies on the dI/dV data.

Reviewer #2 (Remarks to the Author):

[Note from the Editor: Reviewer #2 was asked to look also over the response given to reviewer #3]

The revised paper shows improved dI/dV spectra, where the assignment of the LUMO and HOMO levels seems reasonable. However, the assignment of the levels in the HOMO-LUMO gap is still not conclusive. The comments of referee 2 and referee 3 were not included in the manuscript. The assignment of the levels in the HOMO-LUMO gap to flat band levels seems not really justified. As pointed out before, these states can be explained by the confinement of the surface states, which shifts the levels upwards.

There is no justification to talk about flat bands of the Kagome molecular lattice. The authors showed that the maxima of these levels are in the pores not on the molecules, which is again a clear indication that these states are simply confined surface states from the metallic surface. Therefore, I recommend to revise the manuscript by avoiding the claim of observation of flat bands. It seems conclusive, that confined surface states are observed. In conclusion, I recommend to revise the manuscript.

Point-to-point reply to comments from Reviewer 1:

General Comments: *The authors have satisfactorily addressed my critical issues/questions. The present version manuscript and SI have substantially improved in quality and clarity. However, I am not fully convinced by the authors' argument on the wired bare Au(111) spectrum in previous Fig. 5d (now Fig. 5a). Obviously one could see some disputable oscillations around Fermi. Comparing to the amplitude of the SS around -450 mV, the amplitude of these oscillations (two humps and two dips) are considerably large, which cannot be neglect at all. I assume the overall tip states in this data set is disputable/incorrect, which, as I commented last time, has inevitably compromised all the other dI/dV spectra (and mapping) data. It is suggested that the authors look for other data set with better tip states (if any), to eliminate possible controversies on the dI/dV data.*

Reply to General Comments: We thank the Reviewer for finding that his/her critical issues/questions have been satisfactorily addressed, and our revised manuscript and SI have been substantially improved.

The Reviewer had a question about the bare Au(111) spectrum. We agree with the Reviewer that some oscillations around Fermi level should be further clarified.

About the oscillations (two humps and two dips) appearing in the normalized $(dI/dV)/(I/V)$ spectra in previous **Fig. 5d** (now **Fig. 5a**), the features in the bare Au(111) spectrum around Fermi level might be induced during the normalization. As shown in the raw dI/dV spectra of **Reply Fig. 1a (attached below)**, the gray spectrum of the bare Au(111) surface only displays much weaker oscillations around E_F compared to the intensity of the SSM at around -450 mV, which can be some uncertain tip conditions. These weaker oscillations are enhanced by artificial discontinuous features, as marked by red arrows below and above E_F in different curves of **Reply Fig. 1c**, which result from zero values of the current $I(V)$ spectra (**Reply Fig. 1b**) in the denominator of the normalization. Commonly, to avoid confusion, segments near E_F in the normalized $(dI/dV)/(I/V)$ spectra are omitted, just as shown in the two representative cases in **Reply Fig. 2 (attached below)**.

In addition, following the Reviewer's suggestion, we have looked for another set of data taken from a different superlattice island and using a different tip, shown as a **new Supplementary Fig. 10 (attached below)**. It is seen that while the spectrum from the bare Au(111) does not show the oscillations around the Fermi level (**Supplementary Fig. 10c**), the main features of the labeled peaks of HOMO, FB1, FB2, LUMO, and LUMO+2 in the spectra are in good agreement with the observed ones in **Fig. 5a**. Such observations suggest that the labeled peaks of HOMO, FB1, FB2, LUMO, and LUMO+2 in the spectra are tip-independent, further confirming the correctness of our data and assignments.

In the revision, following commonly similar treatments in normalized dI/dV, to avoid confusion, we have cut the segments near E_F in the **revised Fig. 5a** (attached below in **Reply Fig. 1d**). Two sentences have also been added in the caption of Fig. 5a to clarify this point, reading as "In **a**, the narrow region near E_F (zero bias) is not shown, due to some uncertain features induced during the normalization. Another set of data recorded using a different tip is also shown in Supplementary Fig. 10 for comparison, which gives the tip-independent features in the spectra and maps that are in good agreement with those in **a-g**."

Reply Fig. 1

Figure 3

Ref: Singh, J. P. et al. Retardation of oxidation in Co nanocolumns: Scanning tunneling microscopy study. Appl. Phys. Lett. 81, 4601–4603 (2002).

Figure 2b,d

Ref: Ma, C. et al. Engineering Edge States of Graphene Nanoribbons for Narrow-Band Photoluminescence. ACS Nano 14, 5090–5098 (2020).

Reply Fig. 2

New Supplementary Fig. 10

Point-to-point reply to comments from Reviewer 2:

General Comments: *The revised paper shows improved dI/dV spectra, where the assignment of the LUMO and HOMO levels seems reasonable. However, the assignment of the levels in the HOMO-LUMO gap is still not conclusive. The comments of referee 2 and referee 3 were not included in the manuscript. The assignment of the levels in the HOMO-LUMO gap to flat band levels seems not really justified. As pointed out before, these states can be explained by the confinement of the surface states, which shifts the levels upwards.*

There is no justification to talk about flat bands of the Kagome molecular lattice. The authors showed that the maxima of these levels are in the pores not on the molecules, which is again a clear indication that these states are simply confined surface states from the metallic surface. Therefore, I recommend to revise the manuscript by avoiding the claim of observation of flat bands. It seems conclusive, that confined surface states are observed. In conclusion, I recommend to revise the manuscript.

Reply to General Comments: We thank the Reviewer for finding the improvement of our dI/dV spectra and for their good suggestion. We also apologize that in the previous revision the comments of referee 2 and referee 3 about the assignment of flat bands have not been well addressed.

The Reviewer is correct that the observed levels in the HOMO–LUMO gaps are originated from the confined Shockley surface states. Our strategy is to confine Shockley surface states using designed patterned potential landscapes, as we have highlighted in abstract. We noted that, also as Reviewer mentioned “*As pointed out before, these states can be explained by the confinement of the surface states, which shifts the levels upwards.*”, such previous assignment of *the levels upward shifts* might not capture the core of underlying physics of this observation.

In our study, we find that the maps corresponding to the labeled peaks of FB and FB’ (**Fig. 4**) and FB1 and FB2 (**Fig. 5** and **Supplementary Fig. 10**) give the kagome lattices in real space, by comparing them with the simulated electrostatic potentials (**Fig. 3**). Even though we agree that they are still from the confined surface states accompanying their upward shifts, it is noticed that such quite large upward shifts in energy strongly suggest that some strong interactions are involved. In fact, our finding of the existing kagome lattices at these levels does reveal the underlying physics of their shifts.

To address the comments of the Reviewers, we have revised our manuscript by adding the analysis and discussion.

On Page 9, “According to the DFT simulations (Supplementary Fig. 4), the bandgap of the benzene molecule should be around 5 eV, with the highest occupied molecular orbital (HOMO) and the lowest unoccupied molecular orbital (LUMO) at -2.29 and 2.67 eV, respectively, which suggests that the observed intense peak at 1.46 eV should not originate from the orbitals of the benzene molecule. The dI/dV map acquired at 1.35 V, corresponding to the rising edge of the observed peak to increase the contrast between different sites⁵¹ (Supplementary Fig. 3), displays the kagome lattice with enhanced LDOSs at the exposed Ag surface (Fig. 4c and Supplementary Fig. 5), which is consistent with the simulated electrostatic potential (Fig. 3a). This observation indicates that the observed state results from the confined Shockley SS. In contrast, there is no contribution from the confined Shockley SS (seen by the dim triangles) at a slightly lower energy of 1.00 eV (Fig. 4b). Hence, the pronounced peak centred at 1.46 eV can be reasonably assigned to the kagome flat band from the confined Shockley SS. It is noted that similar levels from the confined Shockley SSs were observed before, but they were simply attributed to their upward shifts^{44,46}. We believe that the much large energy shifts of these states in our experiments should reveal the origin because of the strong interaction in the kagome lattice. The observable peak intensity on the benzene molecule results from the penetration

of confined electrons due to the small size of the benzene molecule. This may reduce the confinement of the patterned surface states and broaden the emerging kagome FB bandwidth, which can be improved by adopting larger building blocks, as shown in the following. Nevertheless, this proof-of-concept result confirms the feasibility of our design strategy.”

On Page 15, “We also need to admit that both our strategy and the tip-assisted manipulation approaches suffer from the broadening effect of the confined electronic states and the possibly emerging kagome flat bands due to the surface-bulk scattering of electrons that results in the limited coherent lifetime on the metallic surface^{3,61}, as observed in our experiment with finite peak widths ranging from about 0.2 to 0.4 eV and in previous reports with similar values^{41,47.}”

REVIEWERS' COMMENTS

Reviewer #1 (Remarks to the Author):

The authors have satisfactorily addressed my criticism on the spectra. The new data set provided in supplementary information looks relatively good. Publication of the current version manuscript is recommended.

Reviewer #2 (Remarks to the Author):

The authors have improved the manuscript. The comparison with the tight binding calculations is convincing. Especially, the ones in the SI-part, where edge states are observed. Therefore, I recommend to publish the work in its present form.